# On The Representation Properties Of The Perturb-Softmax And The Perturb-Argmax Probability Distributions

## Abstract

The Gumbel-Softmax probability distribution allows learning discrete tokens in generative learning, whereas the Gumbel-Argmax probability distribution is useful in learning discrete structures in discriminative learning. Despite the efforts invested in optimizing these models, their properties are underexplored. In this work, we investigate their representation properties and determine for which families of parameters these probability distributions are complete, that is, can represent any probability distribution, and minimal, i.e., can represent a probability distribution uniquely. We rely on convexity and differentiability to determine these conditions and extend this framework to general probability models, denoted Perturb-Softmax and Perturb-Argmax. We conclude the analysis by identifying two sets of parameters that satisfy these assumptions and thus admit a complete and minimal representation. A faster convergence rate of Gaussian-Softmax in comparison to Gumbel-Softmax further motivates our study, as the experimental evaluation validates.

## 1 Introduction

Learning over discrete probabilistic models is an active research field with numerous applications. Examples include learning probabilistic latent representations of semantic categories (Rolfe, 2017) and beliefs (Mnih & Gregor, 2014; Salakhutdinov & Hinton, 2009). The Gumbel-Argmax and Gumbel-Softmax probability distributions are widely applied in machine learning to model and analyze such discrete probability distributions.

The Gumbel-Argmax is an equivalent representation of the softmax operation and plays a key role in the "Follow The Perturb-Leader" (FTPL) family of algorithms in online learning (Hannan, 1957; Kalai & Vempala, 2002; 2005; Rakhlin et al., 2012). Its extension to Gaussian-Argmax allows better bounds on their gradients and consequently provides better regret bounds in linear and high-dimensional settings (Abernethy et al., 2014; 2016; Cohen & Hazan, 2015). The argmax operation allows for efficient sampling, making the Perturb-Argmax probability models pivotal in discriminative learning algorithms of high-dimensional discrete structures (Berthet et al., 2020; Pogančić et al., 2020; Song et al., 2016; Cohen Indelman & Hazan, 2021; Niculae et al., 2018). The Gumbel-Softmax (or the Concrete distribution) probability distribution, which replaces the argmax operation with a softmax operation is easier to optimize and therefore plays a key role in generative learning models (Jang et al., 2017; Maddison et al., 2017; Kusner & Hernández-Lobato, 2016; Ramesh et al., 2021a). The discrete nature of these probability models provides a natural representation of concepts, e.g., in zero-shot text-to-image generation (Ramesh et al., 2021b). While the Gumbel-Argmax and Gumbel-Softmax probability distributions are widely applied in machine learning, their representation properties are still underexplored.

In this work, we investigate the representation properties of the Gumbel-Argmax and Gumbel-Softmax probability distributions. We aim to determine for which families of parameters these distributions are complete and minimal. A distribution is complete if it can represent any probability distribution, and minimal if it can uniquely represent a probability distribution. A complete and minimal distribution is also defined as identifiable. We identify the conditions that determine their representation properties by investigating the Gumbel-Argmax and the Gumbel-Softmax probability

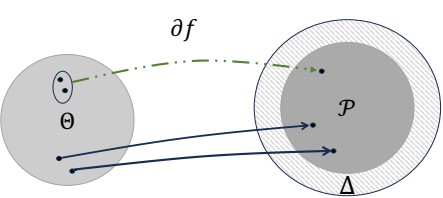 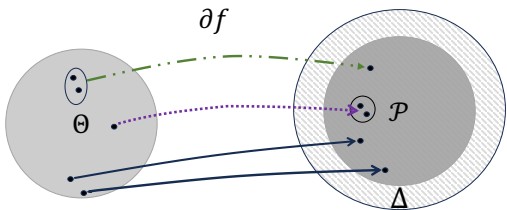

(a) An illustration of the representation properties of the Perturb-Softmax. Its representation is complete for any perturbation distribution under the conditions in Theorem 4.1. The mapping is one-to-one whenever the representation parameters are not linearly constrained (full mapping), and single-valued otherwise (green dashed mapping).

(b) An illustration of the representation properties of the Perturb-Argmax. Its representation is complete under the conditions in Theorem 5.1. It is not identifiable whenever the perturbations follow a discrete distribution (dotted purple mapping), and identifiable for smooth pdf perturbation distributions (green dashed mapping). Moreover, under the conditions in Theorem 5.3, the mapping is one-to-one whenever the representation parameters are not linearly constrained (full mapping).

Figure 1: Illustration of the representation properties of the Perturb-Softmax and the Perturb-Argmax.

distributions as gradients of respective convex functions over their set of parameters. The generality of our framework allows establishing these properties for any random perturbation, for example, Gaussian-Softmax or Gaussian-Argmax, or more broadly, Perturb-Softmax or Perturb-Argmax. Our investigation is further motivated by practical reasons due to the differences in convergence rates between the Gumbel-Softmax and the Gaussian-Softmax, following the theory of measure concentration.

We begin by introducing the notation relating parameters and the relevant probability distributions in Section 3. Subsequently, we investigate the Perturb-Softmax probability models as gradients of the expected log-sum-exp convex function and prove their completeness by connecting their gradients to the relative interior of the probability simplex. In Section 4, we determine the minimality of the Perturb-Softmax by the strict convexity of the expected log-sum-exp when restricted to the respective parameter space. We then investigate the Perturb-Argmax probability models as sub-gradients of the expected-max convex function and establish the conditions for which their parameter space is complete and minimal, see Section 5. Finally, we empirically demonstrate the qualities of Perturb-Softmax extension in generative and discriminative learning settings, showing improved convergence of Gaussian-Softmax over Gumbel-Softmax beyond the linear high-dimensional setting that was investigated in online learning Abernethy et al. (2014); Cohen & Hazan (2015).

Our findings are illustrated in Figure 1 and summarized in Table 1.

Table 1: A summary of the representation properties of the Perturb-Softmax and Perturb-Argmax distributions for perturbation distribution choices.

| Perturbation Dist. | Perturb-Softmax | | | Perturb-Argmax | | |
|---|---|---|---|---|---|---|
| | Completeness | Identifiability | Minimality | Completeness | Identifiability | Minimality |
| Smooth unbounded | | | | | ✓ | ✓* |
| Smooth bounded | ✓† | ✓ | ✓* | ✓‡ | ✗ | ✗ |
| Discrete | | | | | ✗ | ✗ |

† Under the conditions of Theorem 4.1. ‡ Under the conditions of Theorem 5.1. * The case that the representation parameters are not linearly constrained.

## 2 RELATED WORK

The exponential family realized by the softmax operation over its parameters is extensively used in machine learning. However, sampling high-dimensional models is challenging due to its normalizing factor (Geman & Geman, 1984; Goldberg & Jerrum, 2007). The Gumbel-Argmax probability distribution measures the stability of the argmax operation over Gumbel random variables. It is an equivalent representation of softmax operation, thereby enabling efficient sampling from the exponential family (Gumbel, 1954; Luce, 1959). In the context of machine learning, the Gumbel-

Argmax probability models underlie the FTPL online learning algorithm (Hannan, 1957; Kalai & Vempala, 2002; 2005). The Gaussian-Argmax probability models extend the FTPL family of online algorithms and improve their regret bounds (Rakhlin et al., 2012; Abernethy et al., 2014; 2016; Cohen & Hazan, 2015). Our work complements these studies by exploring the properties of Gumbel-Argmax and Gaussian-Argmax. We prove the conditions under which Perturb-Argmax probability models are complete, making them suitable for use in machine learning, and when they are minimal, they can uniquely identify a probability distribution from Perturb-Argmax probability models.

The Gumbel-Softmax probability models were introduced as an alternative to the exponential family and its Gumbel-Argmax equivalent in generative learning (Maddison et al., 2017; Jang et al., 2017). This alternative allows for efficient sampling, making it highly effective for learning with stochastic gradient methods. The discrete nature of the Gumbel-Softmax sampling has been utilized to tokenize the visual vocabulary in the celebrated zero-shot text-to-image generation, DALL-E (Ramesh et al., 2021b). The Gaussian-Softmax probability models were introduced in variational auto-encoders as their closed-form KL-divergence makes it easier to realize as regularization (Potapczynski et al., 2020). Our work extends these works and sets the properties of Gumbel-Softmax and Gaussian-Softmax. We also show that Gaussian-Softmax enjoys faster convergence as the Gaussian distribution decays faster than the Gumbel distribution when approaching infinity.

Berthet et al. (2020) introduced a framework for optimizing discrete problems based on Perturb-Argmax probability models. This framework applies to discriminative learning using the Fenchel-Young losses and relies on convexity to propagate gradients over discrete choices. Similar to our approach, this work adopts a general view and utilizes convexity to explore the gradient properties of Perturb-Argmax models. Our work differs in that we use convexity and differentiability to investigate their representation properties, specifically when these models are complete and minimal. Other methods include blackbox differentiation based on gradients of a surrogate linearized loss (Pogančić et al., 2020), the direct loss minimization technique (Hazan et al., 2010; Keshet et al., 2011; Song et al., 2016; Cohen Indelman & Hazan, 2021) based on gradients of the expected discrete loss, and entropy regularization techniques (Niculae et al., 2018; Martins & Astudillo, 2016). Unlike these methods, we focus on studying the properties of randomized discrete probability models rather than on optimization frameworks.

## 3 PRELIMINARIES

We denote by $\Delta$ the probability simplex, i.e., the set of all probabilities over $d$ discrete events, namely $\Delta \triangleq \{p \in \mathbb{R}^d : p(i) \geq 0, \sum_{i=1}^d p(i) = 1\}$. A parameterized discrete probability distribution $p_\theta(i) \in \Delta$ is determined by its parameters $\theta \in \Theta^d$ that reside in the Euclidean space $\Theta^d \subseteq \mathbb{R}^d$.

### 3.1 COMPLETENESS AND MINIMALITY OF THE SOFTMAX OPERATION

The softmax operation $\mathrm{softmax} : \mathbb{R}^d \to \Delta$ is the standard mapping from the set of parameters $\Theta$ to the probability simplex $\Delta$. Formally, we define $p_\theta^{sm}$ by softmax relation

$$p_\theta^{sm} \triangleq \mathrm{softmax}(\theta) \triangleq \left( \frac{e^{\theta_1}}{\sum_{j=1}^d e^{\theta_j}}, ..., \frac{e^{\theta_d}}{\sum_{j=1}^d e^{\theta_j}} \right). \tag{1}$$

A parameterized family of distributions $\mathbb{P}_\Theta \triangleq \{p_\theta : \theta \in \Theta\}$ is called complete if for every $p \in \Delta$ there exists $\theta \in \Theta$ such that $p_\theta = p$. Alternatively, the mapping from the parameters to their probabilities is onto the probability simplex (surjective). Similarly, a parameterized family of distributions is called minimal, if there is one-to-one mapping between its parameters and their corresponding probability distributions (injective). Formally, $p_\theta \neq p_\tau$ if and only if $\theta \neq \tau$. A complete and minimal mapping is also identifiable, i.e., for every probability $p \in \Delta$ one can identify the unique parameters $\theta \in \Theta$ for which $p = p_\theta$.

The identifiability of the softmax mapping was explored in the context of the exponential family of distributions (Wainwright & Jordan, 2008). One can verify that the set of parameters $\Theta = \mathbb{R}^d$ is complete: for every $p \in \Delta$, one can set $\theta = \log p$, for which $p = softmax(\theta)$. However, the set $\Theta = \mathbb{R}^d$ is not minimal, as the parameter vectors $\theta$ and $\theta + c1$ both realize the same probability,

i.e., $\mathrm{softmax}(\theta) = \mathrm{softmax}(\theta + c\mathbf{1})$. Conversely, a set of parameters $\Theta$ is minimal for the softmax mapping if there are no $\theta, \tau \in \Theta$ for which $\theta_i = \tau_i = c$ for every $i = 1, ..., d$.[1] Consequently, identifiable sets of parameters for the softmax operation can be $\Theta = \{\theta \in \mathbb{R}^d : \sum_j \theta_j = 0\}$ or $\Theta = \{\theta \in \mathbb{R}^d : \theta_1 = 0\}$. In Appendix 8.2.1 we prove that these sets are both complete and minimal.

## 3.2 GUMBEL-SOFTMAX AND GUMBEL-ARGMAX PROBABILITY DISTRIBUTIONS

The Gumbel-Softmax probability distribution emerged as a smooth approximation of the Gumbel-Argmax representation of $p_\theta^{sm}$. We turn to describe the Gumbel-Argmax and Gumbel-Softmax discrete probability distributions.

The Gumbel distribution is a continuous distribution whose probability density function is $\hat{g}(t) = e^{-e^{-(t+c)}}$, where $c \approx 0.5772$ is the Euler-Mascheroni constant. We denote by $\gamma = (\gamma_1, ..., \gamma_d)$ the vector of $d$ independent random variables that follow the Gumbel distribution law and by $g(\gamma) = \prod_{i=1}^d \hat{g}(\gamma_i)$ the probability density function (pdf) of the independent Gumbel distribution.

We denote by $p_\theta^{gam}$ the Gumbel-Argmax probability distribution, which relies on a one-hot representation of the maximal argument. The indicator function $1[condition]$ equals one when the condition holds and zero otherwise. Then, the Gumbel-Argmax probability distribution takes the form:

$$p_\theta^{gam} \triangleq \mathbb{E}_{\gamma \sim g}[\arg\max(\theta + \gamma)] \tag{2}$$

$$p_\theta^{gam}(i) \triangleq \mathbb{E}_{\gamma \sim g}[i = \arg\max(\theta + \gamma)] \triangleq \int_{\mathbb{R}^d} g(\gamma)1[i = \arg\max(\theta + \gamma)]d\gamma \tag{3}$$

Unfortunately, the argmax operation is non-smooth and requires special treatment when used in learning its parameters using gradient methods.

The Gumbel-Softmax probability distribution $p_\theta^{gsm}(i)$ was developed as a smooth approximation of its Gumbel-Argmax counterpart:

$$p_\theta^{gsm} \triangleq \int_{\mathbb{R}^d} g(\gamma)\mathrm{softmax}(\theta + \gamma)d\gamma \triangleq \mathbb{E}_{\gamma \sim g}[\mathrm{softmax}(\theta + \gamma)] \tag{4}$$

We note that we define $p_\theta^{gsm}$ as a $d$-th dimensional vector, and the integral with respect to $\gamma$ (or expectation) is taken with respect to each coordinate of the softmax operation. One can verify that the Gumbel-Softmax is indeed a probability distribution .

The fundamental theorem of extreme value statistics asserts the equivalence between the softmax distribution in Equation 4 and the Gumbel-Argmax distribution in Equation 2, namely $p_\theta^{gsm} = p_\theta^{gam}$, cf. Gumbel (1954). Therefore, the representation properties of completeness and minimality of the softmax operation are identical to the properties of the Gumbel-Argmax probability distribution.

## 3.3 DIFFERENTIBILITY PROPERTIES OF CONVEX FUNCTIONS

We investigate the representation properties of the Gumbel-Softmax and Gumbel-Argmax probability models when they result from gradients of multivariate functions over their set of parameters $\Theta$.

The softmax function is the gradient of the log-sum-exp function:

$$\mathrm{softmax}(\theta) = \nabla \log\left(\sum_{i=1}^d e^{\theta_i}\right). \tag{5}$$

As shown in Section 3.1 this gradient mapping is complete and minimal over a convex subsetset $\Theta \subset \mathbb{R}^d$. Extending this argument to Gumbel-Softmax requires notions of convexity, covered in Appendix 8.1.1. The (sub)differential of convex functions $f(\theta)$ is used in our study as a (multi-valued) mapping between the convex set of the primal domain $\Theta$ and its dual domain $\mathcal{P}$, which is the Gumbel-Softmax or the Gumbel-Argmax probability model. We define the conditions for which:

---

[1]If $\mathrm{softmax}(\theta) = \mathrm{softmax}(\tau)$, and $p_\theta = \mathrm{softmax}(\theta)$, $p_\tau = \mathrm{softmax}(\tau)$, then $p_\theta(i)/p_\tau(i) = 1$ for every $i$, and $\log p_\theta(i) - \log p_\tau(i) = 0$. From the softmax mapping, this translates to $\theta_i - \tau_i = c$ for every $i$ while $c = \log(\sum_j e^{\theta_j}) - \log(\sum_j e^{\tau_j})$.

I $\partial f$ is a single-valued mapping, i.e., $\partial f = \nabla f$. In this case, for every $\theta \in \Theta$ matches a single $\nabla f(\theta) \in \mathcal{P}$. If this property does not hold then the parameters $\theta$ that generate a probability $p$ are not identifiable under this mapping.

II The gradient mapping $\nabla f(\theta)$ is onto the probability simplex $\Delta$. In this case, the set of parameters $\Theta$ is complete, i.e., it can represent (and learn) any probability $p \in \Delta$ using its gradients $\nabla f(\theta)$.

III The gradient mapping $\nabla f(\theta)$ is one-to-one. In this case, the set of parameters $\Theta$ is minimal, i.e., there are no two parameters $\theta, \tau$ that represent the same probability distribution $p \in \mathcal{P}$.

Our framework allows for establishing these relations for any random perturbations, which we refer to as Perturb-Softmax and Perturb-Argmax.

### 3.4 CONVERGENCE RATES

Learning with Perturb-Softmax or Perturb-Argmax involves minimizing the parameters $\theta$ of a function, averaged over the perturbation random variables $\gamma$. Formally, it is of the form: $\min_\theta \mathbb{E}_{x \sim D}[\mathbb{E}_{\gamma \sim g}[f(\theta, x, \gamma)]]$. We denote by $D$ the data distribution, while we consider a sampled training set implicitly in this notation. The expected value $\mathbb{E}_{\gamma \sim g}[f(\theta, x, \gamma)]$ (or its gradient) is approximated by a sample. It is known that Gaussian distribution $\gamma \sim N(0, I)$ enjoys fast convergence (at the rate of $e^{-t^2}$) for functions $f(\cdot)$ with bounded gradient norm. For clarity, we state this phenomena while using $f(\gamma)$ and omitting $x, \theta$:

**Theorem 3.1** (Gaussian Concentration. Van Handel (2014), Theorem 3.25). *Let $\gamma \in \mathbb{R}^d$ be independent Gaussian random variables with zero mean and unit variance. Then*

$$\mathbb{P}[f(\gamma) - \mathbb{E}[f(\gamma)] \geq t] \leq e^{-t^2/2}$$

*for all $t > 0$, where $\|\nabla f(\gamma)\|^2 \leq \sigma^2$. In fact, $f(\gamma)$ is $\sigma^2$-subgaussian.*

On the other hand, the convergence of the exponential family for functions with bounded gradient norm has an exponential rate (of the form $e^{-t}$).

**Theorem 3.2** (Gumbel Concentration. Hazan et al. (2019), Corollary 12). *Let $\gamma \in \mathbb{R}^d$ be independent Gumbel randoms variables with zero mean. Then*

$$\mathbb{P}[f(\gamma) - \mathbb{E}[f(\gamma)] \geq t] \leq e^{-min\{\frac{t}{4\sigma}, \frac{t^2}{32\sigma^2}\}}$$

*for all $t > 0$ where $\|\nabla f(\gamma)\|^2 \leq \sigma^2$ and $\sigma < 1$.*

This strike difference implies that a faster convergence of Gaussian-Softmax in comparison to the Gumbel-Softmax is to be expected, which further motivates our analysis for practical implications.

## 4 PERTURB-SOFTMAX PROBABILITY DISTRIBUTIONS

In this section, we explore the statistical representation properties of the Perturb-Softmax operation as a generalization of the Gumbel-Softmax operation. Our exploration emerges from the connection between the softmax operation and the log-sum-exp convex function, as described in Equation 5. We establish a similar relation between perturb-log-sum-exp and perturb-softmax:

$$f(\theta) = \mathbb{E}_\gamma \left[ \log \left( \sum_{i=1}^d e^{\theta_i + \gamma_i} \right) \right] \qquad (6)$$

$$\nabla f(\theta) = \mathbb{E}_\gamma[\text{softmax}(\theta + \gamma)] \qquad (7)$$

The function $f(\theta)$ is defined for any random perturbation $\gamma$, whether $\gamma$ values are from a discrete, a bounded, or an unbounded set [2]. The function $f(\theta)$ is differentiable since it is the expectation of the differentiable log-sum-exp function and $\nabla f(\theta)$ is attained by the Leibniz rule for differentiation under the integral sign[3]. Also, the function $f(\theta)$ is convex, as it is an expectation of convex log-sum-exp

---

[2]Formally, for unbounded random perturbations $\gamma$ we restrict ourselves to probability density functions for which $f(\theta) < \infty$.

[3]Formally, $\nabla f(\theta)$ is finite whenever the dominant convergence theorem holds. For unbounded $\gamma$ this holds for any probability density function $p(\gamma)$ for which $\lim_{\gamma \to \infty} p(\gamma) \log \left( \sum_{i=1}^d e^{\theta_i + \gamma_i} \right) = 0$. This happens for Gumbel, Gaussian, and other standard probability density functions.

functions. We exploit the convexity of $f(\theta)$ to define the conditions on the parameter space $\Theta$ for which the Perturb-Softmax probability distributions span the probability simplex.

The gradient $\nabla f(\theta)$ maps parameters $\theta$ to a probability, as the softmax vector $\text{softmax}(\theta + \gamma)$ sums up to unity for any $\gamma$ and therefore also in expectation over $\gamma$ (Corollary 8.2). In the next theorem, we determine the conditions for which the gradient mapping spans the relative interior of the probability simplex, i.e., the set of all possible positive probabilities.

**Theorem 4.1** (Completeness of Perturb-Softmax). *Let $\Theta \subseteq \mathbb{R}^d$ be a convex set and let $\gamma = (\gamma_1, ..., \gamma_d)$ be a vector of random variables whose cumulative distribution decays to zero as $\gamma$ approaches $\pm\infty$. Let $h_i(\theta) = \theta_i - max_{j \neq i}\theta_j$ be a continuous function over $\Theta$. If $h_i(\theta)$ are unbounded then $\Theta$ is a complete representation of the Perturb-Softmax probability models:*

$$ri(\Delta) \subseteq \mathbb{E}_\gamma[\text{softmax}(\theta + \gamma)] \subseteq \Delta \tag{8}$$

*Proof.* The proof relies on fundamental notions of the conjugate dual function $f^*(p)$ and its convex domain $\mathcal{P}$, cf. Equations (18, 19) in the Appendix. We begin by considering $f(\theta) = \mathbb{E}_\gamma[\log(\sum_{i=1}^d e^{\theta_i + \gamma_i})]$ and its gradient, which is the Perturb-Softmax model $\nabla f(\theta) = \mathbb{E}_\gamma[\text{softmax}(\theta + \gamma)]$. Equation 22 implies that its gradients, i.e., the Perturb-Softmax probability distributions, reside in their convex domain $\mathcal{P}$ (cf. Equation 21), and contain its relative interior, thus:

$$ri(\mathcal{P}) \subseteq \{\mathbb{E}_\gamma[\text{softmax}(\theta + \gamma)] : \theta \in \Theta\} \subseteq \mathcal{P} \tag{9}$$

To conclude the proof, we prove in Appendix 8.3 that the zero-one probability vectors reside in the closure of $\mathcal{P}$. Since the closure of $\mathcal{P}$ is a convex set, we conclude that it is the probability simplex. □

The above theorem implies that the set of all Perturb-Softmax probability distribution is an almost convex set that resides within the convex set of all probabilities $\Delta$ and contains its relative interior, i.e., the convex set of all positive probabilities $ri(\Delta)$.

Next, we describe the conditions for which the parameter space $\Theta$ is minimal. In this case, two different parameters $\theta \neq \tau$ result in two different Perturb-Softmax models $\mathbb{E}_\gamma[\text{softmax}(\theta + \gamma)] \neq \mathbb{E}_\gamma[\text{softmax}(\tau + \gamma)]$. Interestingly, minimality is tightly tied to strict convexity. We begin by proving that $f(\theta) = \mathbb{E}_\gamma[\log(\sum_{i=1}^d e^{\theta_i + \gamma_i})]$ is strictly convex when restricted to $\Theta$.

**Lemma 4.2** (Strict convexity). *Let $\Theta \subseteq \mathbb{R}^d$ be a convex set and let $\gamma = (\gamma_1, ..., \gamma_d)$ be a vector of random variables and let $f(\theta) = \mathbb{E}_\gamma[\log(\sum_{i=1}^d e^{\theta_i + \gamma_i})]$. If $\Theta$ has no two vectors $\theta \neq \tau \in \Theta$ that are affine translations of each other, for which $\theta_i = \tau_i + c$ for every $i = 1, ..., d$ and some constant $c$ then $f(\theta)$ is strictly convex over $\Theta$, i.e., for any $\theta \neq \tau \in \Theta$ and any $0 < \lambda < 1$ it holds that*

$$f(\lambda\theta + (1 - \lambda)\tau) < \lambda f(\theta) + (1 - \lambda)f(\tau). \tag{10}$$

The proof, based on Hölder's inequality, is provided in Appendix 8.4. The condition that $\theta$ and $\tau$ are not a translation of each other guarantees strict convexity. If $\tau_i = \theta_i + c$ for every $i$, then $f(\tau) = f(\theta) + c$. This linear relation implies that the convexity condition holds with equality.

The minimality theorem is a direct consequence of Lemma 4.2, as strict convexity of differentiable function implies the gradient mapping is one-to-one (cf. Rockafellar (1970), Theorem 26.1).

**Theorem 4.3** (Minimality of Perturb-Softmax). *Let $\Theta \subseteq \mathbb{R}^d$ be a convex set and let $\gamma = (\gamma_1, ..., \gamma_d)$ be a vector of random variables. $\Theta$ is a minimal representation of the Perturb-Softmax probability models if there are no two parameter vectors $\theta \neq \tau \in \Theta$ that are affine translations of each other, for which $\theta_i = \tau_i + c$ for every $i = 1, ..., d$ and some constant c.*

*Proof.* Lemma 4.2 implies that $f(\theta)$ is strictly convex and Equation (7) implies it is differentiable. Recall the conjugate dual function and its domain (Equations (18, 19) in the Appendix) and its gradient mapping $\nabla f : \Theta \to \mathcal{P}$. Since $f(\theta)$ is strictly convex then the function $g(\theta) = \langle p, \theta \rangle - f(\theta)$ is strictly concave hence its maximal argument $\theta^*$ is unique. The gradient vanishes at the maximal argument, $\nabla g(\theta^*) = 0$ or equivalently, $p = \nabla f(\theta^*)$. Since $\theta^*$ is unique then $p$ is unique as well. Therefore $\nabla f(\theta)$ is a one-to-one mapping. □

To conclude, one can use any convex set $\Theta \subset \mathbb{R}^d$ that satisfies the conditions of Theorem 4.1 and Theorem 4.3. Similarly to the softmax probability model, $\Theta$ that is both complete is minimal

can be $\Theta = \{\theta \in \mathbb{R}^d : \sum_j \theta_j = 0\}$ or $\Theta = \{\theta \in \mathbb{R}^d : \theta_1 = 0\}$. Further, this framework can be naturally extended to include temperature scaling, such that the perturb-log-sum-exp with temperature $f_t(\theta) = \mathbb{E}_\gamma \left[ \log \left( \sum_{i=1}^d e^{(\theta_i + \gamma_i)/t} \right) \right]$ is related to the perturb-softmax with temperature:

$$\nabla f_t(\theta) = \mathbb{E}_\gamma[\text{softmax}((\theta + \gamma)/t)], \tag{11}$$

when $t$ is a non-negative temperature hyperparameter.

## 5 PERTURB-ARGMAX PROBABILITY DISTRIBUTIONS

In this section, we explore the statistical representation properties of the Perturb-Argmax operation as a generalization of the Gumbel-Argmax operation. Throughout our investigation, we treat the Perturb-Argmax probability model as the sub-gradient of the expected Perturb-Max function, which we prove in Corollary 9.1 in the Appendix for completeness:

$$f(\theta) \;=\; \mathbb{E}_\gamma \left[ \max_i \{\theta_i + \gamma_i\} \right] \tag{12}$$

$$\partial f(\theta) \;=\; \mathbb{E}_\gamma[\arg\max(\theta + \gamma)] \tag{13}$$

Different than the softmax operation, the argmax operation is not continuous everywhere. This difference arises from the fact that unlike the differentiable log-sum-exp function, the max function is not everywhere differentiable. However, since it is a convex function, its sub-gradient always exists. In the following, we prove that the sub-gradients span the set of all positive probability distributions, i.e., the relative interior of the probability simplex.

**Theorem 5.1** (Completeness of Perturb-Argmax). *Let $\Theta \subseteq \mathbb{R}^d$ be a convex set and let $\gamma = (\gamma_1, ..., \gamma_d)$ be a vector of random variables. Then, $\Theta$ is a complete representation of the Perturb-Argmax probability models:*

$$ri(\Delta) \subseteq \mathbb{E}_\gamma[\arg\max(\theta + \gamma)] \subseteq \Delta \tag{14}$$

*Proof.* The proof technique follows the argument of Theorem 4.1. Given the conditions on $h_i(\theta)$, we can construct a series $\{\theta^{(n)}\}_{n=1}^\infty$ for which $h_i(\theta^{(n)}) = n$ for every $n \in \mathbb{N}$. To conclude the proof, we prove in Appendix 9.1 that $\mathbb{E}_\gamma[\arg\max(\theta^{(n)} + \gamma)]$ approaches the zero-one probability vector as $n \to \infty$. This proves that the zero-one distributions are limit points of probabilities in $\mathcal{P}$, i.e., $cl(\mathcal{P}) = \Delta$. $\square$

The above theorem holds for any type of random perturbation $\gamma$. Next, we show that the statistical properties of the Perturb-Argmax probability models depend on their perturbation type. The minimality of the representation of Perturb-Argmax probability models holds for non-discrete random perturbation $\gamma$. It relies on the differentiability properties of its probability density function $p(\gamma)$.

**Lemma 5.2** (Differentiability of Perturb-Max). *Let $\gamma = (\gamma_1, ..., \gamma_d)$ be a vector of random variables with differentiable probability density function $p(\gamma) = \prod_{i=1}^d p_i(\gamma_i)$ and let $f(\theta) = \mathbb{E}_\gamma[\max\{\theta + \gamma\}]$. Then, $f(\theta)$ is differentiable and its gradient is*

$$\nabla f(\theta) = \mathbb{E}_\gamma[\arg\max(\theta + \gamma)] \tag{15}$$

The proof is provided in Appendix 9.2. Lemma 5.2 shows a single-valued mapping from the parameter space to the probability space. In the following, we show that this mapping brings forth a minimal representation of the Perturb-Argmax probability models under certain conditions.

**Theorem 5.3** (Minimality of Perturb-Argmax). *Let $\Theta \subseteq \mathbb{R}^d$ be a convex set and let $\gamma = (\gamma_1, ..., \gamma_d)$ be a vector of random variables whose probability density functions $p_i(\gamma_i)$ are differentiable and positive. $\Theta$ is a minimal representation of the Perturb-Argmax probability models if there are no two parameter vectors $\theta \neq \tau \in \Theta$ that are affine translations of each other, for which $\theta_i = \tau_i + c$ for every $i = 1, ..., d$ and some constant $c$.*

The proof is provided in Appendix 9.3. It is based on showing that under these conditions, the function $f(\theta) = \mathbb{E}_\gamma[\max\{\theta + \gamma\}]$ is strictly convex. We rely on its one-dimensional function $g(\lambda) \triangleq f(\theta + \lambda v)$ and show that $g''(\lambda) > 0$. Since the function $g(\lambda)$ is convex then $g''(\lambda) \geq 0$, and

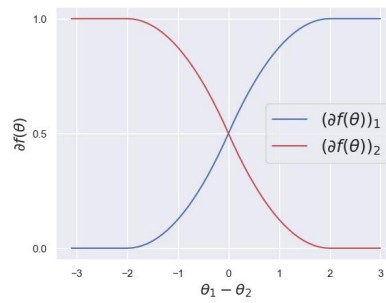 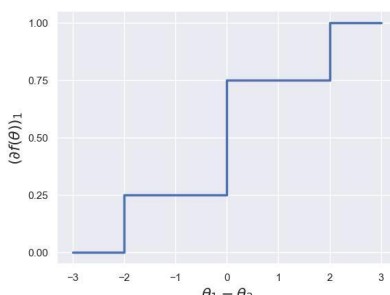

Figure 2: An illustration of $\partial f(\theta)$ for perturbations with a smooth bounded pdf $\gamma \sim U(-1, 1)$. $\partial f(\theta)$ is a single-valued mapping between the parameters and the Perturb-Argmax probability.

Figure 3: An illustration of the sub-differential of $f(\theta)$ w.r.t. $\theta_1$ for discrete random variables $\gamma_i \in \{1, -1\}$ that are uniformly distributed. Notably, the Perturb-Argmax probability is a multi-valued mapping in its overlapping segments, e.g., for $\theta_1 = \theta_2$,

it is enough to show that $g'(\lambda)$ depends on $\lambda$, for which it follows that $g''(\lambda) \neq 0$ and consequently $g''(\lambda) > 0$.

Our theorem conditions require the probability density function to be positive, to ensure that the second derivative is positive as it always accounts for a change in the perturbation space.

### 5.1 NON-MINIMAL REPRESENTATION FOR BOUNDED PERTURBATIONS

In the following, we analyze an example of a Perturb-Argmax distribution when the probability density function of the perturbation is differentiable almost everywhere but bounded. In this case, one can construct a non-minimal representation.

**Proposition 5.4.** *Let $\theta \in \mathbb{R}^2$, and consider i.i.d. random variables with a smooth bounded probability density function $\gamma \sim U(1, -1)$. A single-valued mapping exists between $f(\theta)$ and the Perturb-Argmax probability distribution. However, a one-to-one mapping does not exist.*

*Proof.* The perturb-max function $f(\theta)$ can be expressed as

$$f(\theta) = \theta_2 + \mathbb{E}_{\gamma_1, \gamma_2} \left[ \max\{\theta_1 - \theta_2 + \gamma_1 - \gamma_2, 0\} \right], \tag{16}$$

when the distribution of $\gamma$ is omitted for brevity. We can express $f(\theta)$ by the pdf of the random variable $\gamma_1 - \gamma_2$ (Equation 103 in the appendix). Since $f(\theta)$ is a smooth function, a single-valued mapping exists (Theorem 26.1 Rockafellar (1970)). However, $f(\theta)$ is not strictly convex, hence a one-to-one mapping does not exist and it can be concluded that $\Theta$ is not a minimal representation of the Perturb-Argmax probability. The derivatives of $f(\theta)$, corresponding to the probabilities of $\arg\max$, are illustrated in Figure 2. We defer all details to Section 9.5 in the appendix.

$\square$

### 5.2 DISCRETE PERTURBATIONS AND IDENTIFIABLITY

We next analyze the identifiability of the Perturb-Argmax distribution representation. Since the max function is not always differentiable, the Perturb-Max function $f(\theta)$ (Equation 12) is not always differentiable. However, since the max function is convex, its sub-differential $\partial f(\theta)$ exists. Unfortunately, the function $\partial f(\theta)$ is a multi-valued function, i.e., for some parameters $\theta \in \Theta$ there exist $p \neq q \in \mathcal{P}$ that both are its sub-gradient. Thus, the probability cannot be identified from the parameters when the perturbation is discrete. This property is demonstrated in the following proposition:

**Proposition 5.5.** *Let $\Theta = \mathbb{R}^2$ and $\gamma = (\gamma_1, \gamma_2)$ be a vector of discrete random variables $\gamma_i \in \{1, -1\}$ that are uniformly distributed: $\mathbb{P}[\gamma_i = 1] = \mathbb{P}[\gamma_i = -1] = \frac{1}{2}$. Then, the Perturb-Argmax probability distribution is not identifiable.*

The proof is provided in Appendix 9.4, and it is based on computing the function $f(\theta) = \mathbb{E}_\gamma[\max\{\theta + \gamma\}]$ analytically by taking the expectation w.r.t. $\gamma$. The function $f(\theta)$, illustrated in Figure 6 in the appendix, is continuous and differentiable almost everywhere. However, in its overlapping segments, the function is not differentiable, i.e., it has a sub-differential $\partial f(\theta)$ which is a set of sub-gradients. To prove that the Perturb-Argmax probability model is unidentifiable, we show that $\partial f(\theta)$ is a multi-valued mapping when $\theta_1 = \theta_2$. The sub-differential mapping is illustrated in Figure 3.

## 6 EXPERIMENTS

In this section, we demonstrate the advantage of the Gaussian-Softmax over the commonly used Gumbel-Softmax. Experiments in density estimation and variational inference exhibit that, compared to the Gumbel-Softmax, the Gaussian-Softmax enjoys a faster convergence rate and better approximate discrete distributions.

### 6.1 APPROXIMATING DISCRETE DISTRIBUTIONS

We compare the Gumbel-Softmax and the Normal-Softmax in approximating discrete distributions. The $L_1$ objective function is minimized between the Perturb-Softmax function applied between the fitted and the target discrete distribution pdf, when the latter is denoted by $p_0$. Following the experiment in Potapczynski et al. (2020), we consider two target discrete distributions with finite support: a binomial distribution with parameters $n = 12, p = 0.3$, and a discrete distribution with $p = (\frac{10}{68}, \frac{3}{68}, \frac{4}{68}, \frac{5}{68}, \frac{10}{68}, \frac{10}{68}, \frac{3}{68}, \frac{4}{68}, \frac{5}{68}, \frac{10}{68})$. Figures 4 and 9a show that the Normal-Softmax better approximates both distributions and exhibits faster convergence than the Gumbel-Softmax. We also consider discrete distributions with countably infinite support: a Poisson distribution with $\lambda = 50$, and a negative binomial distribution with $r = 50, p = 0.6$. The results show similar benefits to those with discrete distributions with finite support (Figures 9b, 9c). Moreover, results show that Normal-Softmax has similar benefits to the Invertible Gaussian Reparameterization (IGR) method (Potapczynski et al., 2020), however, unlike our method, the interpretability of the parameters is lost with the IGR method.

Table 2 shows the mean and standard deviation of the $L_1$ objective corresponding to the target discrete distributions of the Gumbel-Softmax and the Normal-Softmax models after 300 iterations, computed over the dimension $d$ of the fitted models. The approximation based on the Normal-Softmax probability model achieves better results in all cases. See more details in Appendix 10.1.

Table 2: $L_1$ mean and standard deviation between the target and approximated probability density function of various target discrete distributions of the Normal-Softmax and Gumbel-Softmax probability models. The best results are in bold.

| Target Distribution | Normal-Softmax | Gumbel-Softmax |
|---|---|---|
| Discrete | **0.026 $\pm$ 0.002** | 0.027$\pm$0.002 |
| Binomial | **0.036$\pm$0.003** | 0.177 $\pm$0.014 |
| Poisson | **0.090$\pm$0.001** | 0.618 $\pm$0.007 |
| Negative Binomial | **0.083 $\pm$ 0.001** | 0.417 $\pm$ 0.004 |

### 6.2 VARIATIONAL INFERENCE

We compared the training ELBO-based loss of categorical Variational-Autoencoders for $N = 10$ variables, each is a $K$-dimensional categorical variable, $K \in [10, 30, 50]$ on the binarized MNIST LeCun & Cortes (2005), the Fashion-MNIST (Xiao et al., 2017), and the Omniglot (Lake et al., 2015) datasets for different smooth perturbation distributions. The architecture consists of an encoder of $X \to FC(300) \to \text{ReLU} \to N * K$, and a matching decoder $N * K \to FC(300) \to \text{ReLU} \to X$. The loss is the traditional composition of the reconstruction error and the KL divergence. See more details in Appendix 10.2

A fair comparison between Perturb-Softmax models with different perturbation distributions requires temperature selection for each model. The temperature is a hyperparameter that affects the models'

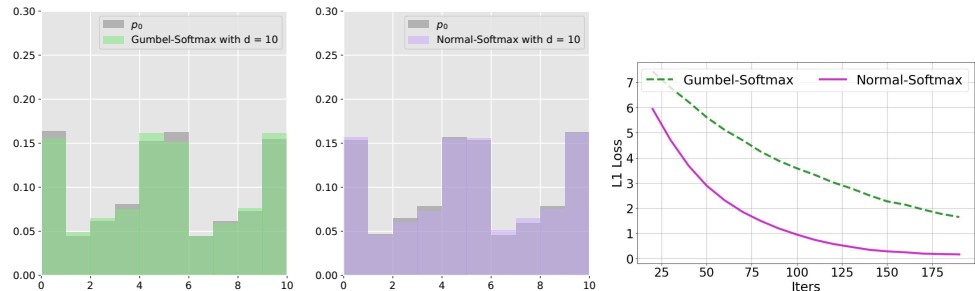

Figure 4: Gumbel-Softmax and Normal-Softmax approximation with dimension d=10 of a target discrete distribution $p_0$. The $L1$ objective over learning iterations is depicted on the right.

performance and, thus should be chosen with cross-validation. We compare the loss obtained by these Pertub-Softmax with temperature model (refer to Equation 11) models for a range of temperatures = $\{0.01, 0.03, 0.07, 0.1, 0.25, 0.4, 0.5, 0.67, 0.85, 1.0\}$. The test set loss is calculated for each temperature with the model achieving the lowest loss on the validation set. Results show that by comparing the best-performing temperature-based models, the Normal-Softmax model consistently achieves the lowest test set loss for all datasets. The results are summarized in Table 3 in the Appendix.

Next, we analyze the training convergence when propagating gradients with the Normal-Softmax or the Gumbel-Softmax of these models for temperature equals 1. Results show that the former achieves better and faster learning convergence in all experiments (Figures 5 and 10 in the appendix).

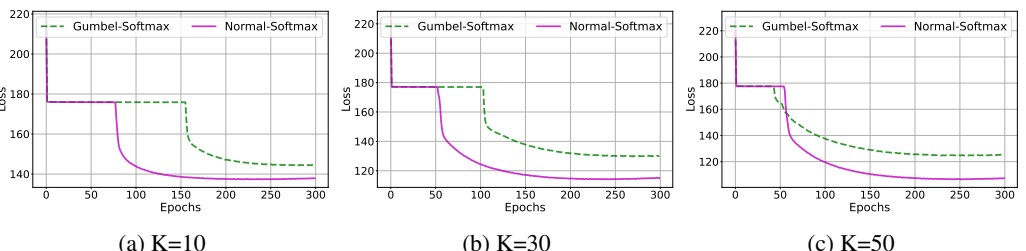

Figure 5: Categorical VAE with Perturb-Softmax training loss on the Omniglot dataset with a $K$-dimensional categorical variable, $K \in [10, 30, 50]$.

## 7    CONCLUSIONS AND LIMITATIONS

Our main contribution is a theoretical study of the representation properties of Gumbel-Softmax and Gumbel-Argmax probability models. Our results provide a theoretical justification that for fitting probabilities one need not limit oneself to Gumbel random perturbations, as the representation properties with any smooth probability density perturbation distribution are the same. In particular, the Gumbel-Softmax distribution has no representation benefits over the Normal-Softmax distribution. Moreover, we show that the representation of the Perturb-Argmax probability distribution is unidentifiable whenever the perturbations follow a discrete distribution, hence one may fail in fitting probabilities from parameters since the Perturb-Argmax probabilities are multi-valued mappings from parameters. Though our analysis shows that Gaussian-Softmax and Gaussian-Argmax share the same representation properties, the Gaussian distribution enjoys faster convergence in probabilistic estimation. These advantages are demonstrated in our experiments.

As our framework extends the Gumbel-Softmax and Gumbel-Argmax probability models, it suffers the same limitations as these models. Specifically, these models cannot be efficiently applied to probability spaces for which the softmax function is too expensive to be computed. Further, our investigation does not easily apply to high-dimensional probability distributions (e.g., the Gumbel-Sinkhorn distribution that manifests in learning latent matching representations (Mena et al., 2018)), and the investigation of such distributions' properties is left for future work.

ETHICS STATEMENT

In this study, we revisit the widely applied discrete probability models and study their representation characteristics to understand their properties better. To our knowledge, this study has no negative implications or applications.

REPRODUCIBILITY STATEMENT

To promote the reproducibility of the results in this paper, we took the following measures: 1. Experiments are based on previous research, and the experiment settings, optimization, and hyperparameters are detailed in the main text and appendix. 2. Code will be made public. 3. The statistics of results for approximating discrete distributions are reported. 4. The theoretical results are illustrated (refer to Figures 2, 3) and complete proofs are detailed in the main text and the appendix.

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

# 8 APPENDIX

## 8.1 RELATED WORK

### 8.1.1 CONVEXITY

We consider convex functions $f : \Theta \to \mathbb{R}$ over its convex domain $\Theta = dom(f)$ and follow the notation in Rockafellar (1970). A function over a domain $\Theta$ is convex if for every $\theta, \tau \in \Theta$ and $0 \leq \lambda \leq 1$ it holds that $f(\lambda\theta + (1 - \lambda)\tau) \leq \lambda f(\theta) + (1 - \lambda)f(\tau)$. A function is strictly convex if for every $\theta \neq \tau \in \Theta$ and $0 < \lambda < 1$, it holds that $f(\lambda\theta + (1 - \lambda)\tau) < \lambda f(\theta) + (1 - \lambda)f(\tau)$. Whenever $f(\theta)$ is twice differentiable, a function is strictly convex if its Hessian is positive definite.

Convexity is a one-dimensional property. A function $f : \Theta \to \mathbb{R}$ is convex if and only if its one-dimensional reduction $g(\lambda) \triangleq f(\theta + \lambda v)$ is convex in every admissible direction $v$, i.e., whenever $\theta, \theta + \lambda v \in \Theta$. A twice differentiable function $f(\theta)$ is strictly convex if the second derivative of $g(\lambda)$ is positive in every admissible direction $v$. In this case, we denote $g'(\lambda)$ by $\nabla_v f(\theta)$ and call it a directional derivative:

$$\nabla_v f(\theta) \triangleq \lim_{\epsilon \to 0} \frac{f(\theta + \epsilon v) - f(\theta)}{\epsilon} \tag{17}$$

A multivariate function is differentiable if its directional derivative is the same in every direction $v \in \mathbb{R}^d$, namely $\nabla f(\theta) = \nabla_v f(\theta)$ for every $v \in \mathbb{R}^d$.

Convexity admits duality correspondence. Any primal convex function $f(\theta)$ has a dual conjugate function $f^*(p)$

$$f^*(p) = \max_{\theta \in \Theta} \{\langle p, \theta \rangle - f(\theta)\} \tag{18}$$

$$\mathcal{P} \triangleq dom(f^*) \triangleq \{p : f^*(p) < \infty\} \tag{19}$$

Since $f^*(p)$ is a convex function, its domain $\mathcal{P}$ is a convex set.

A sub-gradient $p \in \partial f(\theta)$ satisfies $f(\tau) \geq f(\theta) + \langle p, \tau - \theta \rangle$ for every $\tau \in \Theta$. The sub-gradient is intimately connected to directional derivatives. Theorem 23.2 in Rockafellar (1970) states that

$$p \in \partial f(\theta) \quad \text{iff} \quad \nabla_v f(\theta) \geq \langle p, v \rangle, \ \forall \text{ admissible } v. \tag{20}$$

The vector $v$ is admissible if $\theta + \epsilon v \in \Theta$ for small enough $\epsilon$.

The set of all sub-gradients is called sub-differential at $\theta$ and is denoted by $\partial f(\theta)$. A convex function is differentiable at $\theta$ when $\partial f(\theta)$ consists of a single vector, and it is denoted by $\nabla f(\theta)$. The sub-differential is a multi-valued mapping between the primal parameters and dual parameters, i.e.,

$$\partial f : \Theta \to \mathcal{P} \tag{21}$$

One can establish with this property the definition of sub-gradient at the optimal point $\theta^* = \arg\max_\theta \{\langle p, \theta \rangle - f(\theta)\}$. In this case, $0 \in \partial (\langle p, \theta^* \rangle - f(\theta^*))$, where the sub-gradient is taken with respect to $\theta$ at the maximal argument $\theta^*$. From the linearity of the sub-gradient, there holds: $p \in \partial f(\theta^*)$, or equivalently, $\{\partial f(\theta) : \theta \in \Theta\} \subseteq \mathcal{P}$. Using the connection between sub-gradients and directional derivatives, one can show that whenever directional derivatives exist, one can infer a sub-gradient, i.e., the set of all sub-gradients contains the relative interior of $\mathbb{P}$, cf. Theorem 23.4 Rockafellar (1970):

$$ri(\mathcal{P}) \subseteq \{\partial f(\theta) : \theta \in \Theta\} \subseteq \mathcal{P} \tag{22}$$

## 8.2 PERTURB-SOFTMAX PROBABILITY DISTRIBUTIONS

### 8.2.1 COMPLETENESS AND MINIMALITY OF THE SOFTMAX OPERATION

**Theorem 8.1.** *The representation of the softmax distribution defined over $\theta \in \mathbb{R}^d$ is complete. It is minimal when the corresponding log-sum-exp*

$$f(\theta) = \log(\sum_{i=1}^{d} e^{\theta_i}) \tag{23}$$

*is a strictly convex function (A paraphrase of Wainwright & Jordan (2008), Proposition 3.1).*

*Proof.* First, we note that the derivatives of $f(\theta)$ (Eq. 23),

$$\frac{\partial f(\theta)}{\partial \theta_j} = \text{softmax}(\theta), \tag{24}$$

correspond to the softmax probabilities $p_\theta^{sm}$ (Eq. 1). $p_\theta^{sm}$ is a Gibbs model, hence the representation is complete.

Let $\theta, \tau \in \mathbb{R}^d$ and denote $e^{\theta_i} = u_i$, $e^{\tau_i} = v_i$. Then, for $\lambda \in (0,1)$

$$f(\lambda\theta + (1-\lambda)\tau) = \log(\sum_{i=1}^{d} e^{\lambda\theta_i + (1-\lambda)\tau_i}) \tag{25}$$

$$= \log(\sum_{i=1}^{d} u_i^\lambda v_i^{1-\lambda}). \tag{26}$$

Applying Hölder's inequality to Equation 26 :

$$\log\left(\sum_{i=1}^{d} u_i^\lambda v_i^{1-\lambda}\right) \leq \log(\sum_{i=1}^{d} u_i^{\lambda\frac{1}{\lambda}})^\lambda (\sum_{i=1}^{d} v_i^{(1-\lambda)\frac{1}{1-\lambda}})^{1-\lambda}) \tag{27}$$

$$= \lambda\log(\sum_{i=1}^{d} u_i) + (1-\lambda)\log(\sum_{i=1}^{d} v_i) \tag{28}$$

$$= \lambda f(\theta) + (1-\lambda)f(\tau). \tag{29}$$

Therefore, it holds that

$$f(\lambda\theta + (1-\lambda)\tau) \leq \lambda f(\theta) + (1-\lambda)f(\tau), \tag{30}$$

proving that $f(\theta)$ is convex.

Hölder's inequality holds with equality if and only if there exists a constant $c \in \mathbb{R}$ such that

$$|v_i^{1-\lambda}| = c|u_i^\lambda|^{\frac{1}{\lambda}-1} \quad \forall i \tag{31}$$

$$e^{(1-\lambda)\tau_i} = ce^{\lambda\theta_i(\frac{1}{\lambda}-1)} \quad \forall i \tag{32}$$

$$\tau_i = \frac{\log(c)}{1-\lambda} + \theta_i \quad \forall i, \tag{33}$$

in which case $\theta$ and $\tau$ are linearly constrained and there exists some $\alpha \in \mathbb{R}^2, \alpha \neq 0$ such that $\alpha_1\theta + \alpha_2\tau = const$. Therefore, when the representation of $f(\theta)$ is minimal $f(\theta)$ is strictly convex.

Then, consider any $\theta, \tau \in \Theta^d$, such that $\Theta = \{\theta \in \mathbb{R}^d : \theta_1 = 0\}$. Hölder's inequality holds strictly as there can not exist a constant $c$ such that Equation 33 holds for all $i$ if $d > 1$, proving that the representation of $\Theta = \{\theta \in \mathbb{R}^d : \theta_1 = 0\}$ is minimal.

Consider $\Theta = \{\theta \in \mathbb{R}^d : \sum_j \theta_j = 0\}$. Then, let $\theta, \tau \in \mathbb{R}^d : \sum_i \theta_i = 0, \sum_i \tau_i = 0$, and denote $p_i \propto e^{\theta_i}$ and $q_i \propto e^{\tau_i}$. The proof requires showing that if there exists $i : p_i \neq q_i$ such that $\theta_i \neq \tau_i + c$. Equivalently, it requires proving that if it holds that $\theta_i - \tau_i = 0$ for any $i$, then $p_i = q_i$ for all $i$. Explicitly,

$$p_i = \frac{e_i^\theta}{\sum_j e_j^\theta} \tag{34}$$

$$q_i = \frac{e_i^\tau}{\sum_j e_j^\tau}. \tag{35}$$

Then, by marginalization it holds that

$$\log(p_i) = \log(qi) \quad \longleftrightarrow \quad \theta_i - \log(\sum_j e^{\theta_j}) = \tau_i - \log(\sum_j e^{\tau_j}) \tag{36}$$

$$\longleftrightarrow \quad \sum_i \theta_i - \sum_i \log(\sum_j e^{\theta_j}) = \sum_i \tau_i - \sum_i \log(\sum_j e^{\tau_j}) \tag{37}$$

$$\longleftrightarrow \quad d\log(\sum_j e^{\theta_j}) = d\log(\sum_j e^{\tau_j}), \tag{38}$$

which concludes the proof.

$\Theta = \{\theta \in \mathbb{R}^d : \sum_j \theta_j = 0\}$ is complete by the conditions of our completeness theorems by setting $n$ at the $i^{th}$ positions and $-a/d$ everywhere else.

$\Theta = \{\theta \in \mathbb{R}^d : \theta_1 = 0\}$ is complete by the conditions of our completeness theorems by setting $n$ at the $1^{st}$ positions and $0$ everywhere else. $\qquad \square$

**Corollary 8.2.** *The derivative of the expected log-sum-exp $f(\theta)$ (Equation 6) is a probability function.*

*Proof.* Denote $p_{\gamma,j} = \frac{\partial f(\theta)}{\partial \theta_j}$, then

$$\sum_{j=1}^d p_{\gamma,j} \quad = \quad \sum_{j=1}^d \mathbb{E}_\gamma \left[ \frac{e^{\theta_j + \gamma_j}}{\sum_{i=1}^d e^{\theta_i + \gamma_i}} \right] \tag{39}$$

$$= \quad \mathbb{E}_\gamma \left[ \sum_{j=1}^d \frac{e^{\theta_j + \gamma_j}}{\sum_{i=1}^d e^{\theta_i + \gamma_i}} \right] = 1. \tag{40}$$

Also,

$$\mathbb{E}_\gamma \left[ \frac{e^{\theta_j + \gamma_j}}{\sum_{i=1}^d e^{\theta_i + \gamma_i}} \right] \geq 0 \quad \forall j. \tag{41}$$

$\qquad \square$

### 8.3 SUPPORTING PROOF FOR THEOREM 4.1

First, we prove that $f(\theta)$ (Equation 6) is a closed proper convex function and is also essentially smooth. $f(\theta)$ is a convex function as a maximum of convex (linear) functions. Then, $f(\theta)$ is proper as its effective domain is nonempty and it never attains the value $-\infty$, since $\theta \in \mathbb{R}^d$. $f(\theta)$ is infinitely differentiable throughout the domain, therefore it is a smooth function throughout its domain. $f(\theta)$ is a smooth convex function on $\mathbb{R}^d$, therefore it is in particular essentially smooth. The smoothness of $f(\theta)$ guarantees its continuity, and since $\mathbb{R}^d$ can be considered a closed set, then $f(\theta)$ is a closed function.

Given the conditions of the theorem on $h_i(\theta)$, we can construct a series $\{\theta^{(n)}\}_{n=1}^\infty$ for which $h_i(\theta^{(n)}) = n$ for every $n \in \mathbb{N}$. We prove that $\mathbb{E}_\gamma[\arg\max(\theta^{(n)} + \gamma)]$ approaches the zero-one probability vector as $n \to \infty$.

$$\mathbb{E}_\gamma \left[ \frac{e^{\theta_i^{(n)} + \gamma_i}}{\sum_{j=1}^d e^{\theta_j^{(n)} + \gamma_j}} \right] \quad = \mathbb{E}_\gamma \left[ \frac{1}{\sum_{j=1}^d e^{\theta_j^{(n)} + \gamma_j - \theta_i^{(n)} - \gamma_i}} \right] \tag{42}$$

$$= \mathbb{E}_\gamma \left[ \frac{1}{1 + \sum_{j \neq i} e^{\theta_j^{(n)} - \theta_i^{(n)} + \gamma_j - \gamma_i}} \right] \tag{43}$$

$$\geq \mathbb{E}_\gamma \left[ \frac{1}{1 + \sum_{j \neq i} e^{-n + \gamma_j - \gamma_i}} \right] \overset{n \to \infty}{\to} 1 \tag{44}$$

The limit argument holds since the probability of $\gamma_1, ..., \gamma_d$ decay as they tend to infinity. This proves that the zero-one distributions are limit points of probabilities in $\mathcal{P}$, i.e., $cl(\mathcal{P}) = \Delta$.

### 8.4 PROOF OF LEMMA 4.2

In the following, we prove the strict convexity of the expected log-sum-exp of Lemma 4.2.

*Proof.* Let $\theta, \tau \in \mathbb{R}^d$ and $0 < \lambda < 1$. Then

$$f(\lambda\theta + (1-\lambda)\tau) = \mathbb{E}_\gamma \left[ \log\left( \sum_{i=1}^d e^{\lambda(\theta_i + \gamma_i) + (1-\lambda)(\tau_i + \gamma_i)} \right) \right] = \mathbb{E}_\gamma \left[ \log\left( \sum_{i=1}^d u_i v_i \right) \right], \tag{45}$$

where $u_i \triangleq e^{\lambda(\theta_i + \gamma_i)}$ and $v_i \triangleq e^{(1-\lambda)(\tau_i + \gamma_i)}$. Applying Hölder's inequality $\langle u, v \rangle \leq \|v\|_{1/\lambda} \cdot \|u\|_{1/(1-\lambda)}$ we obtain the convexity condition of the log-sum-exp function:

$$f(\lambda\theta + (1-\lambda)\tau) \leq \lambda f(\theta) + (1-\lambda)f(\tau). \tag{46}$$

To prove strict convexity we note that Hölder's inequality for non-negative vectors $u_i, v_i$ holds with equality if and only if there exists a constant $\alpha \in \mathbb{R}$ such that $v_i = \alpha u_i^{\frac{1-\lambda}{\lambda}}$ for every $i = 1, ..., d$, or equivalently:

$$e^{(1-\lambda)(\tau_i + \gamma_i)} = \left(e^{\lambda(\theta_i + \gamma_i)}\right)^{\frac{1-\lambda}{\lambda}} \longleftrightarrow \tau_i = \theta_i + c \tag{47}$$

Where $c = \frac{\log \alpha}{1-\lambda}$. Therefore, if $\tau \neq \theta + c$, then $\langle u, v \rangle < \|v\|_{1/\lambda} \cdot \|u\|_{1/(1-\lambda)}$ for every $\gamma$ and consequently it also holds when applying the logarithm function and taking an expectation with respect to $\gamma$. $\square$

# 9 PERTURB-ARGMAX PROBABILITY DISTRIBUTIONS

**Corollary 9.1.** *We prove that the derivative of the expected maximizer is the probability of the* arg max. *Namely, that* $\frac{\partial}{\partial \theta}\mathbb{E}_{\gamma \sim g}[\max_i\{\theta_i + \gamma_i\}] = P_\gamma(\arg\max_i\{\theta_i + \gamma_i\} = i)$.

*Proof.* First, by differentiating under the integral:

$$\partial\mathbb{E}_\gamma[\max\{\theta + \gamma\}] = \mathbb{E}_\gamma[\partial\max\{\theta + \gamma\}] \tag{48}$$

Writing a subgradient of the max-function using an indicator function (an application of Danskin's Theorem):

$$\partial\max_i\{\theta_i + \gamma_i\} = \mathbb{1}[\arg\max_i(\theta_i + \gamma_i) = i] \tag{49}$$

The proof then follows by applying the expectation to both sides of Equation 49. $\square$

## 9.1 SUPPORTING PROOF FOR THEOREM 5.1

Given the conditions on $h_i(\theta)$, we can construct a series $\{\theta^{(n)}\}_{n=1}^\infty$ for which $h_i(\theta^{(n)}) = n$ for every $n \in \mathbb{N}$. We show that $\mathbb{E}_\gamma[\arg\max(\theta^{(n)} + \gamma)]$ approaches the zero-one probability vector as $n \to \infty$.

$$\mathbb{P}[i = \arg\max(\theta^{(n)} + \gamma) \quad = \mathbb{P}\left[\theta_i^{(n)} + \gamma_i \geq \max_{j \neq i}\{\theta_j^{(n)} + \gamma_j\}\right] \tag{50}$$

$$\geq \mathbb{P}\left[\theta_i^{(n)} + \gamma_i \geq \max_{j \neq i}\{\theta_j^{(n)}\} + \max_{j \neq i}\{\gamma_j\}\right] \tag{51}$$

$$\geq \mathbb{P}\left[\gamma_i \geq -n + \max_{j \neq i}\{\gamma_j\}\right] \stackrel{n \to \infty}{\to} 1$$

The limit argument holds since the probability of $\gamma_1, ..., \gamma_d$ decay as they tend to infinity.

**Corollary 9.2.** *The convex conjugate of* $f(\theta)$ *takes the following values:*

$$f^*(\lambda) = \begin{cases} -\mathbb{E}_{\gamma \sim g}[\gamma_{\hat{i}}] & \text{if } \lambda = \lambda^* \\ \infty & \text{otherwise}, \end{cases} \tag{52}$$

*where* $\gamma_{\hat{i}}$ *denotes* $\gamma_i$ *for which* $\arg\max_i\{\theta_i + \gamma_i\} = \hat{i}$.

*Proof.* Then, the convex conjugate of $f(\theta)$, when $\lambda_i^*$ is denoted by $p_i$ is

$$
\begin{aligned}
f^*\left(\lambda\right) &= \sum_i \theta_i p_i - \mathbb{E}_{\gamma \sim g}\left[\max_i\{\theta_i + \gamma_i\}\right] && (53) \\[2mm]
&= \sum_i \theta_i p_i - \mathbb{E}_\gamma\left[\left(\sum_{\hat{i}} \mathbb{1}_{\arg\max_i\{\theta_i+\gamma_i\}=\hat{i}}\right)\max_i\{\theta_i+\gamma_i\}\right] && (54) \\[2mm]
&= \sum_i \theta_i p_i - \sum_{\hat{i}} \mathbb{E}_\gamma\left[\mathbb{1}_{\arg\max_i\{\theta_i+\gamma_i\}=\hat{i}}\max_i\{\theta_i+\gamma_i\}\right] && (55) \\[2mm]
&= \sum_i \theta_i p_i - \sum_{\hat{i}} \mathbb{E}_\gamma\left[\mathbb{1}_{\arg\max_i\{\theta_i+\gamma_i\}=\hat{i}}(\theta_{\hat{i}} + \gamma_{\hat{i}})\right] && (56) \\[2mm]
&= \sum_i \theta_i p_i - \sum_{\hat{i}} \mathbb{E}_\gamma\left[\mathbb{1}_{\arg\max_i\{\theta_i+\gamma_i\}=\hat{i}}\theta_{\hat{i}}\right] - \sum_{\hat{i}} \mathbb{E}_\gamma\left[\mathbb{1}_{\arg\max_i\{\theta_i+\gamma_i\}=\hat{i}}\gamma_{\hat{i}}\right] && (57) \\[2mm]
&= \sum_i \theta_i p_i - \sum_{\hat{i}} p_{\hat{i}}\theta_{\hat{i}} - \sum_{\hat{i}} \mathbb{E}_\gamma\left[\mathbb{1}_{\arg\max_i\{\theta_i+\gamma_i\}=\hat{i}}\gamma_{\hat{i}}\right] && (58) \\[2mm]
&= -\sum_{\hat{i}} \mathbb{E}_\gamma\left[\mathbb{1}_{\arg\max_i\{\theta_i+\gamma_i\}=\hat{i}}\gamma_{\hat{i}}\right] && (59) \\[2mm]
&= -\sum_{\hat{i}} p_{\hat{i}}\mathbb{E}_\gamma\left[\gamma_{\hat{i}}\right] && (60) \\[2mm]
&= -\mathbb{E}_\gamma\left[\gamma_{\hat{i}}\right] && (61)
\end{aligned}
$$

where $\gamma_{\hat{i}}$ denotes $\gamma_i$ for which $\arg\max_i\{\theta_i + \gamma_i\} = \hat{i}$.

$\square$

### 9.2 PROOF OF LEMMA 5.2

*Proof.* By reparameterization

$$
f(\theta) = \int_{\mathbb{R}^d} p(\gamma)\max\{\theta + \gamma\}d\gamma = \int_{\mathbb{R}^d} p(\theta - \gamma)\max\{\gamma\}d\gamma \tag{62}
$$

The proof concludes by differentiating under the integral sign while noting that $p(\theta - \gamma)$ is differentiable. $\square$

### 9.3 PROOF OF THEOREM 5.3

In what follows we prove the minimality of the Perturb-Argmax of Theorem 5.3.

*Proof.* Similarly to the Perturb-Softmax setting, we prove that under these conditions the function $f(\theta) = \mathbb{E}_\gamma[\max\{\theta + \gamma\}]$ is strictly convex. For this we rely on its one dimensional function $g(\lambda) \triangleq f(\theta + \lambda v)$ and show that $g''(\lambda) > 0$. Since the function $g(\lambda)$ is convex then $g''(\lambda) \geq 0$, and it is enough to show that $g'(\lambda)$ depends on $\lambda$, for which it follows that $g''(\lambda) \neq 0$ and consequently $g''(\lambda) > 0$.

$g'(\lambda)$ is the directional derivative $\nabla_v f(\theta)$ in every admissible direction $v = \tau - \theta$, for $\tau, \theta \in \Theta$. The theorem conditions assert that $v$ is not the constant vector, i.e., $v \neq c\mathbf{1}$, where $c$ is some constant and $\mathbf{1} = (1, ..., 1)$ is the all-one vector.

We assume, without loss of generality, that $\max\{\theta + \gamma\} \triangleq \max_{i=1,...,d}\{\theta_i + \gamma_i\}$ is chosen between two indexes, namely $\max\{\theta + \gamma\} = \max\{\theta_1 + \gamma_1, \theta_j + \gamma_j\}$. This is possible as we treat $j$ to be $j = \arg\max_{i\neq 1}\{\theta_j + \gamma_j\}$. We denote by $p_1(\gamma_1)$ the differentiable probability density function of $\gamma_1$. We denote remaining random variables as $\gamma_{-1} \triangleq (\gamma_2, ..., \gamma_d)$, their probability density function by their measure $d\mu(\gamma_{-1})$.

We analyze $g'(\lambda) = \lim_{\epsilon \to 0} \frac{1}{\epsilon}(g(\lambda + \epsilon) - g(\lambda))$

$$
\begin{aligned}
g(\lambda) &= E_\gamma[\max\{\theta_1 + \lambda v_1 + \gamma_1, \theta_j + \lambda v_j + \gamma_j\}] \\
&= \int d\mu(\gamma_{-1}) \int_{-\infty}^{\alpha(\gamma_j)} p_1(\gamma_1)(\theta_j + \lambda v_j + \gamma_j)d\gamma_1 + \int d\mu(\gamma_{-1}) \int_{\alpha(\gamma_j)}^{\infty} p_1(\gamma_1)(\theta_1 + \lambda v_1 + \gamma_1)d\gamma_1
\end{aligned}
\tag{63}
$$

$\alpha(\gamma_j)$ is the threshold for which $\gamma_1$ shifts the maximal value to $\theta_1 + \lambda v_1 + \gamma_1$, namely $\alpha(\gamma_j) = \theta_j + \lambda v_j + \gamma_j - \theta_1 - \lambda v_1$.

With this notation, $g(\lambda + \epsilon) =$

$$
\int d\mu(\gamma_{-1}) \int_{-\infty}^{\alpha(\gamma_j)+\epsilon(v_j-v_1)} p_1(\gamma_1)(\theta_j + (\lambda+\epsilon)v_j + \gamma_j)d\gamma_1 + \int d\mu(\gamma_{-1}) \int_{\alpha(\gamma_j)+\epsilon(v_j-v_1)}^{\infty} p_1(\gamma_1)(\theta_1 + (\lambda+\epsilon)v_1 + \gamma_1)d\gamma_1
\tag{64}
$$

Their difference is composed of four terms:

$$
\begin{aligned}
g(\lambda + \epsilon) - g(\lambda) &= \\
&\int d\mu(\gamma_{-1}) \int_{-\infty}^{\alpha(\gamma_j)+\epsilon(v_j-v_1)} p_1(\gamma_1)\epsilon v_j d\gamma_1 + \int d\mu(\gamma_{-1}) \int_{\alpha(\gamma_j)+\epsilon(v_j-v_1)}^{\infty} p_1(\gamma_1)\epsilon v_1 d\gamma_1 \\
&+ \int d\mu(\gamma_{-1}) \int_{\alpha(\gamma_j)}^{\alpha(\gamma_j)+\epsilon(v_j-v_1)} p_1(\gamma_1)(\theta_j + (\lambda+\epsilon)v_j + \gamma_j)d\gamma_1 \\
&- \int d\mu(\gamma_{-1}) \int_{\alpha(\gamma_j)}^{\alpha(\gamma_j)+\epsilon(v_j-v_1)} p_1(\gamma_1)(\theta_1 + (\lambda+\epsilon)v_1 + \gamma_1)d\gamma_1
\end{aligned}
$$

Taking the limit to zero $\lim_{\epsilon \to 0} \frac{1}{\epsilon}(g(\lambda + \epsilon) - g(\lambda))$, the last two terms cancel out, since when taking the limit then $\gamma_1 = \alpha(\gamma_j)$ and $\alpha(\gamma_j) = \theta_j + \lambda v_j + \gamma_j - \theta_1 - \lambda v_1$ by definition, or equivalently, $\alpha(\gamma_j) + \theta_1 + \lambda v_1 = \theta_j + \lambda v_j + \gamma_j$. Therefore

$$
g'(\lambda) = \int d\mu(\gamma_{-1}) \int_{-\infty}^{\alpha(\gamma_j)} p_1(\gamma_1)v_j d\gamma_1 + \int d\mu(\gamma_{-1}) \int_{\alpha(\gamma_j)}^{\infty} p_1(\gamma_1)v_1 d\gamma_1
\tag{65}
$$

We conclude that by the conditions of the theorem $\alpha(\gamma_j) = \theta_j + \lambda v_j + \gamma_j - \theta_1 - \lambda v_1$ is a function of $\lambda$ since there exists $j$ for which $v_j - v_1 \neq 0$ and the probability density function $p_1(\gamma_1) > 0$ therefore it assigns mass on the intervals $[-\infty, \alpha(\gamma_j)]$ and $[\alpha(\gamma_j), \infty]$. Therefore $g'(\lambda)$ is non-constant function of $\lambda$ and $g''(\lambda) \neq 0$. $\qquad\square$

### 9.4 PROOF OF PROPOSITION 5.5

Let $\theta \in \mathbb{R}^2$, and consider i.i.d. random variables $\gamma \in \{1, -1\}$, such that $P(\gamma = 1) = P(\gamma = -1) = \frac{1}{2}$. Let $f(\theta) = \mathbb{E}_\gamma[\max_i\{\theta_i + \gamma_i\}]$ denote the expected perturbed maximum over the domain $\mathbb{R}^d$. Let $f(\theta)$ take values over the extended real domain $\mathbb{R} \cup \{\pm\infty\}$. Clearly, $P(1 = \arg\max_i\{\theta_i + \gamma_i\}) = P(\theta_1 + \gamma_1 \geq \theta_2 + \gamma_2)$.

Then, $f(\theta)$ can be explicitly expressed as:

$$
\begin{aligned}
f(\theta) &= \mathbb{E}_\gamma\left[\max_i\{\theta_i + \gamma_i\}\right] \\
&= P(\gamma_1 = 1, \gamma_2 = 1)(1 + \max\{\theta_1, \theta_2\}) + P(\gamma_1 = -1, \gamma_2 = -1)(\max\{\theta_1, \theta_2\} - 1) \\
&+ P(\gamma_1 = 1, \gamma_2 = -1)(\max\{\theta_1 + 1, \theta_2 - 1\}) + P(\gamma_1 = -1, \gamma_2 = 1)(\max\{\theta_1 - 1, \theta_2 + 1\}) \\
&= \frac{1}{2}(\max\{\theta_1, \theta_2\}) + P(\gamma_1 = 1, \gamma_2 = -1)(\max\{\theta_1 + 1, \theta_2 - 1\}) + P(\gamma_1 = -1, \gamma_2 = 1)(\max\{\theta_1 - 1, \theta_2 + 1\}) \\
&= \frac{1}{2}(\max\{\theta_1, \theta_2\}) + \frac{1}{2}\left(\frac{\max\{\theta_1 + 1, \theta_2 - 1\}}{2} + \frac{\max\{\theta_1 - 1, \theta_2 + 1\}}{2}\right)
\end{aligned}
\tag{66}
$$

Equation 67 suggests that $f(\theta)$ takes the following form:

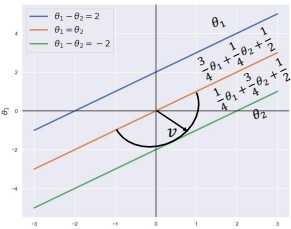

Figure 6: An illustration of the function $f(\theta)$ (Equation 67) and the direction vector $v : v_1 \geq v_2$ at $\theta = (0, 0)$.

$$
f(\theta) = \begin{cases} \theta_1 & \text{if } \theta_1 \geq \theta_2 + 2 \\ \frac{3}{4}\theta_1 + \frac{1}{4}\theta_2 + \frac{1}{2} & \text{if } \theta_2 + 2 \geq \theta_1 \geq \theta_2 \\ \frac{3}{4}\theta_2 + \frac{1}{4}\theta_1 + \frac{1}{2} & \text{if } \theta_2 - 2 \leq \theta_1 \leq \theta_2 \\ \theta_2 & \text{if } \theta_1 \leq \theta_2 - 2 \end{cases} \tag{67}
$$

Recall, that we aim to prove that the Perturb-Argmax probability model is unidentifiable. The function $f(\theta)$, illustrated in Figure 6 in the appendix, is continuous and differentiable almost everywhere. However, in its overlapping segments, i.e., when $\theta_1 = \theta_2 + 2$, $\theta_1 = \theta_2$ and $\theta_1 + 2 = \theta_2$, the function is not differentiable, i.e., it has a sub-differential $\partial f(\theta)$ which is a set of sub-gradients. To prove that the Perturb-Argmax probability model is unidentifiable, we show that $\partial f(\theta)$ is a multi-valued mapping when $\theta_1 = \theta_2$. In particular, we show that every probability distribution $p = (p_1, p_2)$ with $p_1 \in [\frac{1}{4}, \frac{3}{4}]$ satisfies $p \in \partial f(\theta)$.

For this task, we recall the connection between sub-gradients and directional derivatives: $p \in \partial f(\theta)$ if $\nabla_v f(\theta) \geq \langle p, v \rangle$ for every $v \in \mathbb{R}^2$. When $\theta_1 = \theta_2 = c$, then $f(\theta) = c + \frac{1}{2}$, thus for the direction $v = (v_1, v_2)$ for which $v_1 \geq v_2$ holds $\nabla_v f(\theta) = \frac{3}{4}v_1 + \frac{1}{4}v_2$. Recall that $p \in \partial f(\theta)$ if $\frac{3}{4}v_1 + \frac{1}{4}v_2 \geq \langle p, v \rangle$ for every $v_1 \geq v_2$. Thus we conclude that $p$ must satisfy $p_1 \leq \frac{3}{4}$. Since the same holds to $v_2 \geq v_1$ then $p_1 \geq \frac{1}{4}$. Taking both these conditions, $p \in \partial f(\theta)$ when $\frac{1}{4} \leq p_1 \leq \frac{3}{4}$. Therefore, $\partial f(\theta)$ is multi-valued mapping, or equivalently, the parameters $\theta = (\theta_1, \theta_2)$ are not identifying probability distributions. The sub-differential mapping is

$$
(\partial f(\theta))_1 = \begin{cases} 1 & \text{if } \theta_1 > \theta_2 + 2 \\ [\frac{3}{4}, 1] & \text{if } \theta_1 = \theta_2 + 2 \\ \frac{3}{4} & \text{if } \theta_2 + 2 > \theta_1 > \theta_2 \\ [\frac{1}{4}, \frac{3}{4}] & \text{if } \theta_1 = \theta_2 \\ \frac{1}{4} & \text{if } \theta_2 - 2 < \theta_1 < \theta_2 \\ [0, \frac{1}{4}] & \text{if } \theta_1 = \theta_2 - 2 \\ 0 & \text{if } \theta_1 < \theta_2 - 2. \end{cases} \tag{68}
$$

as illustrated in Figure 3.

### 9.5 PROOF OF PROPOSITION 5.4

The perturb-max function $f(\theta)$ can be expressed as

$$
f(\theta) = \mathbb{E}_{\gamma_1, \gamma_2 \sim [-1,1]} \left[ \max_i \{\theta_i + \gamma_i\} \right] \tag{69}
$$

$$
= \mathbb{E}_{\gamma_1, \gamma_2} \left[ \max\{\theta_1 + \gamma_1, \theta_2 + \gamma_2\} \right] - \mathbb{E}_{\gamma_2} [\gamma_2] \tag{70}
$$

$$
= \mathbb{E}_{\gamma_1, \gamma_2} \left[ \max\{\theta_1 + \gamma_1, \theta_2 + \gamma_2\} - \gamma_2 \right] \tag{71}
$$

$$
= \mathbb{E}_{\gamma_1, \gamma_2} \left[ \max\{\theta_1 + \gamma_1 - \gamma_2, \theta_2\} + \theta_2 - \theta_2 \right] \tag{72}
$$

$$
= \theta_2 + \mathbb{E}_{\gamma_1, \gamma_2} \left[ \max\{\theta_1 - \theta_2 + \gamma_1 - \gamma_2, 0\} \right], \tag{73}
$$

when the second equation holds since $E_{\gamma_2 \sim [-1,1]} [\gamma_2] = 0$, and the distribution of $\gamma$ is omitted for brevity.

Define $\theta = \theta_1 - \theta_2$ and $Z = \gamma_1 - \gamma_2$. Then, the random variable $Z$ has a triangular distribution. The random variable $Z$ has the following cdf:

$$F_Z(z) = P(\gamma_1 - \gamma_2 \leq z) \tag{74}$$

$$= \begin{cases} 0 & \text{if } z < -2 \\ \frac{1}{4}\frac{(2+z)^2}{2} & \text{if } 0 > z \geq -2 \\ \frac{1}{4}(4 - \frac{(2-z)^2}{2}) & \text{if } 2 \geq z \geq 0 \\ 0 & \text{if } z > 2 \end{cases} \tag{75}$$

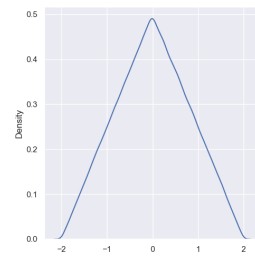

Figure 7: Simulation of the density of the difference between 1M iid uniform(-1,1) random variables.

The random variable $Z$ has the following pdf, also verified in simulation of the density of the difference between 1M iid $U(-1, 1)$ random variables (Figure 7):

$$f_Z(z) = \begin{cases} \frac{1}{4}(2 + z) & \text{if } 0 > z \geq -2 \\ \frac{1}{4}(2 - z) & \text{if } 2 \geq z \geq 0 \\ 0 & \text{otherwise} \end{cases} \tag{76}$$

With the pdf of the random variable $Z$ (Equation 76) consider $f(\theta)$ the appropriate range of $\theta$.

1. **Case:** $-\theta < -2$
   In this case $\theta_1 - \theta_2 > 2$, therefore

$$f(\theta) = \mathbb{E}_{\gamma_1, \gamma_2 \sim [-1,1]} \left[ \max_i \{\theta_i + \gamma_i\} \right] \tag{77}$$

$$= \mathbb{E}_{\gamma_1 \sim [-1,1]} [\theta_1 + \gamma_1] \tag{78}$$

$$= \theta_1 + \mathbb{E}_{\gamma_1 \sim [-1,1]} [\gamma_1] \tag{79}$$

$$= \theta_1 \tag{80}$$

2. **Case:** $0 \leq -\theta \leq 2$

$$f(\theta) = \theta_2 + \mathbb{E}_z [\max\{\theta + z, 0\}] \tag{81}$$

$$= \theta_2 + \int_{-\theta}^{2} f_Z(z)(\theta + z)dz \tag{82}$$

$$= \theta_2 + \int_{-\theta}^{2} \frac{1}{4}(2 - z)(\theta + z)dz \tag{83}$$

$$= \theta_2 + \frac{1}{2}\int_{-\theta}^{2} \theta dz + \frac{1}{2}\int_{-\theta}^{2} z dz - \frac{1}{4}\int_{-\theta}^{2} z\theta dz - \frac{1}{4}\int_{-\theta}^{2} z^2 dz \tag{84}$$

$$= \frac{1}{4}\left(\frac{4}{3} + 2(\theta_1 + \theta_2) + (\theta_1 - \theta_2)^2 + \frac{1}{6}(\theta_1 - \theta_2)^3\right) \tag{85}$$

3. **Case:** $-2 \leq -\theta \leq 0$

$$f(\theta) = \theta_2 + \mathbb{E}_z [\max\{\theta + z, 0\}] \tag{86}$$

$$= \theta_2 + \int_{-\theta}^0 f_Z(z)(\theta + z)dz + \int_0^2 f_Z(z)(\theta + z)dz \tag{87}$$

$$= \theta_2 + \int_{-\theta}^0 \frac{1}{4}(2+z)(\theta + z)dz + \int_0^2 \frac{1}{4}(2-z)(\theta + z)dz \tag{88}$$

$$= \theta_2 + \frac{1}{4}(\theta^2 - \frac{1}{6}\theta^3) + \frac{1}{4}(2\theta + \frac{4}{3}) \tag{89}$$

$$= \frac{1}{4}\left(\frac{4}{3} + 2(\theta_1 + \theta_2) + (\theta_1 - \theta_2)^2 - \frac{1}{6}(\theta_1 - \theta_2)^3\right) \tag{90}$$

$$\int_{-\theta}^0 \frac{1}{4}(2+z)(\theta+z)dz = \frac{1}{2}\int_{-\theta}^0 \theta dz + \frac{1}{2}\int_{-\theta}^0 z dz + \frac{1}{4}\int_{-\theta}^0 z\theta dz + \frac{1}{4}\int_{-\theta}^0 z^2 dz \tag{91}$$

$$= \frac{1}{2}[\theta z]_{-\theta}^0 + \frac{1}{2}\left[\frac{z^2}{2}\right]_{-\theta}^0 + \frac{1}{4}\left[\frac{z^2\theta}{2}\right]_{-\theta}^0 + \frac{1}{4}\left[\frac{z^3}{3}\right]_{-\theta}^0 \tag{92}$$

$$= \frac{1}{4}(\theta^2 - \frac{1}{6}\theta^3) \tag{93}$$

$$\int_0^2 \frac{1}{4}(2-z)(\theta+z)dz = \int_0^2 \frac{1}{4}(2\theta + 2z - z\theta - z^2)dz \tag{94}$$

$$= \frac{1}{2}\int_0^2 \theta dz + \frac{1}{2}\int_0^2 z dz - \frac{1}{4}\int_0^2 z\theta dz - \frac{1}{4}\int_0^2 z^2 dz \tag{95}$$

$$= \frac{1}{2}[\theta z]_0^2 + \frac{1}{2}\left[\frac{z^2}{2}\right]_0^2 - \frac{1}{4}\left[\frac{z^2\theta}{2}\right]_0^2 - \frac{1}{4}\left[\frac{z^3}{3}\right]_0^2 \tag{96}$$

$$= \frac{1}{4}(2\theta + \frac{4}{3}) \tag{97}$$

4. **Case:** $-\theta > 2$

   In this case $\theta_1 - \theta_2 < -2$, therefore

$$f(\theta) = \mathbb{E}_{\gamma_1, \gamma_2 \sim [-1,1]}\left[\max_i \{\theta_i + \gamma_i\}\right] \tag{98}$$

$$= \mathbb{E}_{\gamma_2 \sim [-1,1]}\left[\theta_2 + \gamma_2\right] \tag{99}$$

$$= \theta_2 + \mathbb{E}_{\gamma_2 \sim [-1,1]}\left[\gamma_2\right] \tag{100}$$

$$= \theta_2 \tag{101}$$

To conclude,

$$f(\theta) = \begin{cases} \theta_1 & \text{if } -\theta < -2 \\ \frac{1}{4}\left(\frac{4}{3} + 2(\theta_1 + \theta_2) + (\theta_1 - \theta_2)^2 + \frac{1}{6}(\theta_1 - \theta_2)^3\right) & \text{if } 0 \le -\theta \le 2 \\ \frac{1}{4}\left(\frac{4}{3} + 2(\theta_1 + \theta_2) + (\theta_1 - \theta_2)^2 - \frac{1}{6}(\theta_1 - \theta_2)^3\right) & \text{if } -2 \le -\theta \le 0 \\ \theta_2 & \text{if } -\theta > 2 \end{cases} \tag{102}$$

Alternatively, one writes

$$f(\theta) = \begin{cases} \theta_1 & \text{if } \theta > 2 \\ \frac{1}{4}\left(\frac{4}{3} + 2(\theta_1 + \theta_2) + (\theta_1 - \theta_2)^2 - \frac{1}{6}(\theta_1 - \theta_2)^3\right) & \text{if } 2 \ge \theta \ge 0 \\ \frac{1}{4}\left(\frac{4}{3} + 2(\theta_1 + \theta_2) + (\theta_1 - \theta_2)^2 + \frac{1}{6}(\theta_1 - \theta_2)^3\right) & \text{if } 0 \ge \theta \ge -2 \\ \theta_2 & \text{if } \theta < -2 \end{cases} \tag{103}$$

The derivative of $f(\theta)$ (Equation 102), $\frac{\partial}{\partial \theta}f(\theta) = (\frac{\partial}{\partial \theta_1}f(\theta), \frac{\partial}{\partial \theta_2}f(\theta))$ corresponds to the probabilities of the arg max, $(P_\gamma\left(\arg\max_i\{\theta_i + \gamma_i\} = 1\right), P_\gamma\left(\arg\max_i\{\theta_i + \gamma_i\} = 2\right))$ :

$$\frac{\partial}{\partial\theta}f(\theta) = \begin{cases} (1,0) & \text{if } -\theta < -2 \\ (\frac{1}{2} + \frac{1}{2}\theta + \frac{1}{8}\theta^2, \frac{1}{2} - \frac{1}{2}\theta - \frac{1}{8}\theta^2) & \text{if } 0 \leq -\theta \leq 2 \\ (\frac{1}{2} + \frac{1}{2}\theta - \frac{1}{8}\theta^2, \frac{1}{2} - \frac{1}{2}\theta + \frac{1}{8}\theta^2) & \text{if } -2 \leq -\theta \leq 0 \\ (0,1) & \text{if } -\theta > 2 \end{cases} \tag{104}$$

$$\frac{\partial}{\partial\theta}f(\theta) = \begin{cases} (1,0) & \text{if } \theta > 2 \\ (\frac{1}{2} + \frac{1}{2}\theta - \frac{1}{8}\theta^2, \frac{1}{2} - \frac{1}{2}\theta + \frac{1}{8}\theta^2) & \text{if } 2 \geq \theta \geq 0 \\ (\frac{1}{2} + \frac{1}{2}\theta + \frac{1}{8}\theta^2, \frac{1}{2} - \frac{1}{2}\theta - \frac{1}{8}\theta^2) & \text{if } 0 \geq \theta \geq -2 \\ (0,1) & \text{if } \theta < -2 \end{cases} \tag{105}$$

Then, the partial derivatives $\frac{\partial}{\partial\theta}f(\theta) \in [0,1]$ and sum to 1, as expected.

$$\frac{\partial}{\partial\theta_1}f(\theta) + \frac{\partial}{\partial\theta_2}f(\theta) = 1 \tag{106}$$

1. **Case:** $0 \leq -\theta \leq 2$

$$\frac{\partial}{\partial\theta}f(\theta) = (\frac{1}{4}(2 + 2\theta_1 - 2\theta_2 + \frac{3\theta_1^2 - 6\theta_1\theta_2 + 3\theta_2^2}{6}), \frac{1}{4}(2 + 2\theta_2 - 2\theta_1 + \frac{-3\theta_1^2 + 6\theta_1\theta_2 - 3\theta_2^2}{6})) \tag{107}$$

$$= (\frac{1}{2} + \frac{1}{2}\theta + \frac{1}{8}\theta^2, \frac{1}{2} - \frac{1}{2}\theta - \frac{1}{8}\theta^2) \tag{108}$$

The global minimum of the derivative $f(\theta)$ w.r.t. $\theta_1$, $\min \frac{\partial}{\partial\theta_1}f(\theta) = 0$ for $\theta = -2$, since

$$\frac{\partial \frac{1}{4}(2 + 2\theta_1 - 2\theta_2 + \frac{3\theta_1^2 - 6\theta_1\theta_2 + 3\theta_2^2}{6})}{\partial\theta_1} = \frac{1}{4}(2 + \theta_1 - \theta_2), \tag{109}$$

in which case $\frac{\partial}{\partial\theta_2}f(\theta) = 1$. The global maximum of the derivative $f(\theta)$ w.r.t. $\theta_1$, $\max \frac{\partial}{\partial\theta_1}f(\theta) = \frac{1}{2}$ for $-\theta = 0$, in which case $\frac{\partial}{\partial\theta_2}f(\theta) = \frac{1}{2}$.

The global maximum of the derivative $f(\theta)$ w.r.t. $\theta_2$, $\max \frac{\partial}{\partial\theta_2}f(\theta)) = 1$ for $\theta_2 = \theta_1 + 2$, since

$$\frac{\partial \frac{1}{4}(2 + 2\theta_2 - 2\theta_1 + \frac{-3\theta_1^2 + 6\theta_1\theta_2 - 3\theta_2^2}{6}))}{\partial\theta_2} = \frac{1}{4}(2 + \theta_1 - \theta_2), \tag{110}$$

in which case $\frac{\partial}{\partial\theta_1}f(\theta) = 0$. The global minimum of the derivative $f(\theta)$ w.r.t. $\theta_2$, $\max \frac{\partial}{\partial\theta_2}f(\theta) = \frac{1}{2}$ for $-\theta = 0$, in which case $\frac{\partial}{\partial\theta_1}f(\theta) = \frac{1}{2}$.

## 10 EXPERIMENTS

We use a 16GB 6-Core Intel Core i7 CPU in both experiments.

### 10.1 APPROXIMATING DISCRETE DISTRIBUTIONS

Our experiments are based on the publicly available code of Potapczynski et al. (2020). In all experiments, a thousand samples from a target distribution $p_0$ are sampled to approximate its probability density function parameters, based on the $L_1$ objective function. Optimization is based on the Adam optimizer (Kingma & Ba, 2017) with a learning rate $1.e - 2$. The fitted parameters are initialized uniformly, as visualized in Figure 8. Figure 9 shows results for fitting discrete distributions with countably infinite support: a Poisson distribution with $\lambda = 50$, and a negative binomial distribution with $r = 50, p = 0.6$ respectively. The Normal-Softmax distribution better approximates both distributions and exhibits faster convergence than the Gaussian-Softmax.

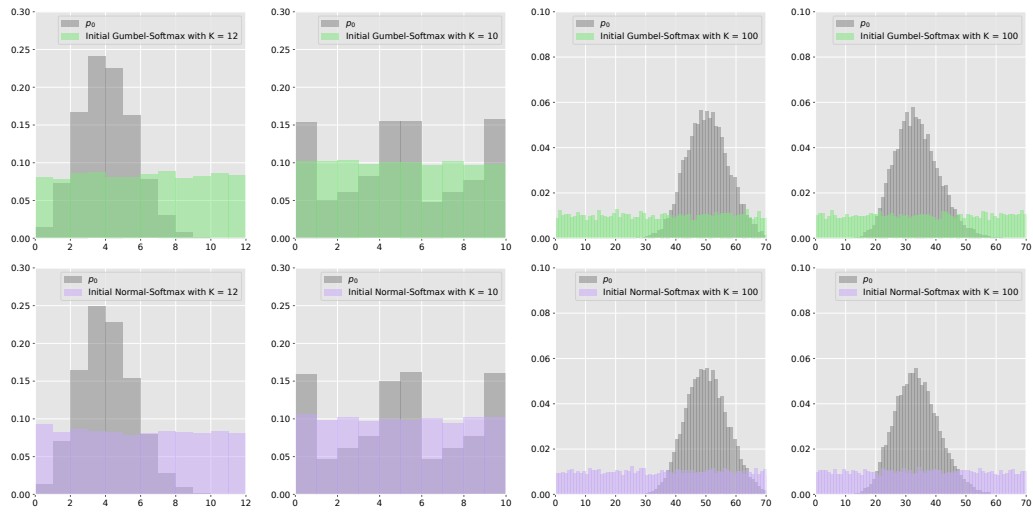

(a) a binomial distribution  (b) a discrete distribution  (c) a Poisson distribution  (d) a negative binomial distribution

Figure 8: Visualization of the uniform initialization of the fitted Gumbel-Softmax (top row) and Normal-Softmax (bottom row) for the target discrete distributions: (8a) a target binomial distribution $p_0$ with $n = 12, p = 0.3$, (8b) a discrete distribution with $p = (\frac{10}{68}, \frac{3}{68}, \frac{4}{68}, \frac{5}{68}, \frac{10}{68}, \frac{10}{68}, \frac{3}{68}, \frac{4}{68}, \frac{5}{68}, \frac{10}{68})$, (8c) a Poisson distribution with $\lambda = 50$, and (8d) a negative binomial distribution with $r = 50, p = 0.6$.

## 10.2 VARIATIONAL INFERENCE

This experiment is based on the publicly available implementation of the Gumbel-Softmax-based implementation of the discrete VAE in Direct-VAE. Optimization is based on the Adam optimizer (Kingma & Ba, 2017) with a learning rate of $1.e-3$. Batch size is set to $100$. We use the regular train/ test splits and follow previous research splits (e.g., as in Lorberbom et al. (2019)). The MNIST and the Fashion-MNIST datasets' training set comprises $60,000$ images, and the test set comprises $10,000$ images. For the Omniglot, the training set comprises $24,345$ images, and the test set comprises $8,070$ images. We consider a range of temperatures = $\{0.01, 0.03, 0.07, 0.1, 0.25, 0.4, 0.5, 0.67, 0.85, 1.0\}$ and select the best-performing models. Comparing the best-performing temperature-based models reveals that the Normal-Perturb models achieve lower test set results on all three datasets (Table 3).

Figure 10 shows the training performance of the variational inference experiment for the Fashion-MNIST dataset (Xiao et al., 2017) with $N = 10$ discrete variables, each is a $K$-dimensional categorical variable, $K \in [10, 30, 50]$, when temperature is set to 1, demonstrating favorable convergence of Normal-Softmax models for all categorical variable dimensions.

| | MNIST dataset | | Fashion- MNIST dataset | | Omniglot dataset | |
|---|---|---|---|---|---|---|
| Temp. | Best test set loss Gumbel-Softmax | Best test set loss Normal-Softmax | Best test set loss Gumbel-Softmax | Best test set loss Normal-Softmax | Best test set loss Gumbel-Softmax | Best test set loss Normal-Softmax |
| 0.01 | 117.29 | 114.26 | 176.68 | 175.94 | 135.87 | 130.77 |
| 0.03 | 109.72 | 107.73 | 158.04 | 153.04 | 132.71 | **130.55** |
| 0.07 | 105.51 | 103.87 | 148.61 | 145.72 | **132.13** | 133.32 |
| 0.10 | 105.79 | 102.89 | 146.53 | 142.73 | 133.6 | 132.52 |
| 0.25 | **104.4** | **101.15** | **144.71** | **141.04** | 137.19 | 233.04 |
| 0.40 | 104.63 | 112.45 | 145.62 | 143.17 | 134.88 | 172.1 |
| 0.50 | 105.94 | 135.16 | 144.89 | 153.05 | 136.34 | 141.29 |
| 0.67 | 123.04 | 186.53 | 149.73 | 170.55 | 139.75 | 132.73 |
| 0.85 | 168.01 | 244.31 | 174.65 | 219.68 | 158.47 | 132.07 |
| 1.00 | 215.42 | 302.01 | 200.74 | 252.86 | 179.59 | 208.08 |

Table 3: Summary of the best test set VAE loss for Normal-Softmax and Gaussian-Softmax model for various temperatures on the MNIST, Fashion-MNIST, and Omniglot datasets. The best-performing temperature-based models are in bold.

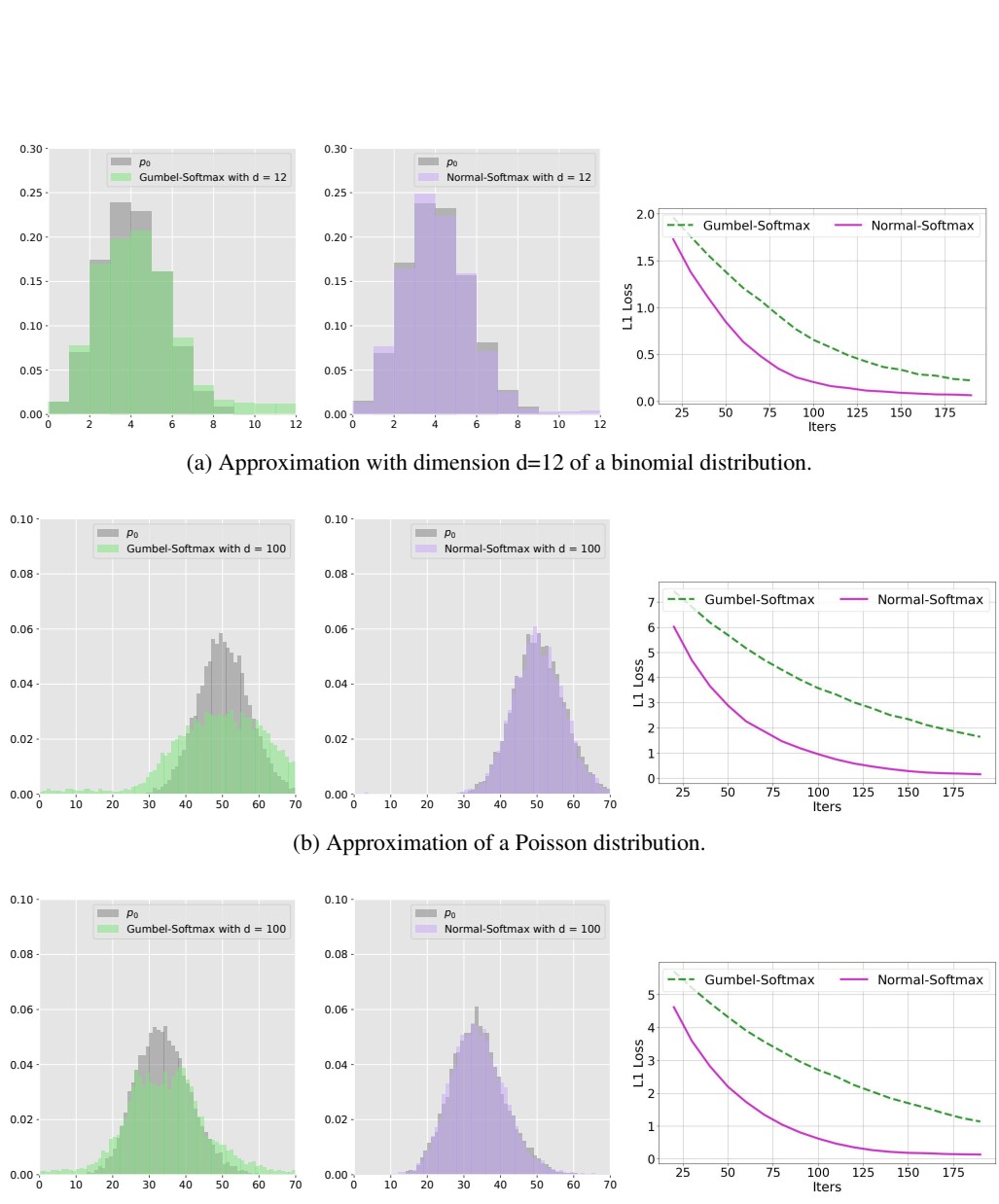

(a) Approximation with dimension d=12 of a binomial distribution.

(b) Approximation of a Poisson distribution.

(c) Approximation of a negative binomial distribution.

Figure 9: Gumbel-Softmax and Normal-Softmax approximation with a finite dimension of target distributions $p_0$. Top row: approximation with dimension $d = 12$ of a binomial distribution (with finite support). Middle and bottom tow: approximation with dimension $d = 100$ of a Poisson and a negative binomial distribution (both with countably infinite support), respectively. The $L1$ objective over learning iterations is depicted on the right.

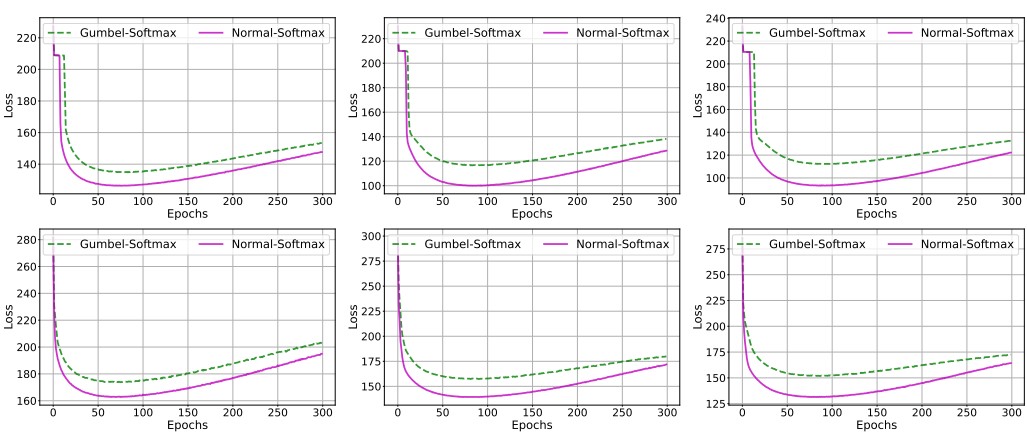

Figure 10: Categorical VAE with Perturb-Softmax training loss on the MNIST (top row), and Fashion-MNIST (bottom row) datasets with a $K$-dimensional categorical variable, $K \in [10, 30, 50]$.

