# OpenReview forum: "On The Representation Properties Of The Perturb-Softmax And The Perturb-Argmax Probability Distributions"
_ICLR.cc/2025/Conference — Submitted to ICLR 2025_

### Official Review · Reviewer_P6qr · 2024-11-03

**Soundness:** 3
**Presentation:** 3
**Contribution:** 3
**Rating:** 6
**Confidence:** 4

**Summary:**

The paper characterizes the the ability of particular parametrized families of probability mass functions (PMFs) to fit an arbitrary PMF on a finite probability space. The main result is a set of conditions under which the Perturbed-Softmax and Perturbed-Argmax families enjoy completeness (whether the parameter-to-distribution map is onto) and/or minimality (whether the parameter-to-distribution map is one-to-one).

**Strengths:**

- The paper is concise and overall well-written (see Weaknesses for feedback). In particular, the proofs in the main text and footnotes are helpful and executed well; usually, this type of presentation is hard to get right.
- The mathematical analysis is elegant and general enough to account for any perturbation distribution.
- The experimental testbed is generally helpful, although I do not believe the provide much insight regarding the theory (see Weaknesses).
- The appendices are helpful for making the paper self-contained overall, but the proof appendices should not be theorem-proof lists. Try to make them self-contained as well; concretely, please reintroduce the theorem, add proof outlines, discuss any techniques that are novel to this paper versus those that are adapted from others with citations, etc)/

**Weaknesses:**

- The earlier part of the presentation could be improved, in that components from the Related Work and Preliminaries could be incorporated into the Introduction. For example, I don't believe the paper mentions one of the major motivations of the Gumbel-Softmax as an approximation to the discrete sampling operation in VAEs that can be backpropagated through. Similarly, while online learning methods are stated as a motivation, the reader should be able to know where the perturbed-argmax/softmax operations appear in their methods, even without having a background in online learning. Equations for the perturbed operations could appear much earlier in the text, so that the Gumbel-softmax/argmax, Gaussian-softmax/argmax, and their perturbed variants are not so abstract.
- Unless I may be misunderstanding, their seem to be some technical errors in the main text proofs. For example, the proof of Theorem 4.1 does not use the $h_i$ function, so it must be incomplete. This assumption is in fact used in Theorem 5.1, but not included in the theorem statement.
- I do not believe the current experiments align well with rest of the paper. In particular, what is shown is that Gaussian perturbations result in a predictor that learns faster with respect to validation loss. However, the rest of the paper is about the representation capabilities of parametrized models, and if I understand correctly, the Gumbel and Gaussian perturbed models have the same representation properties. Moreover, this difference between Gaussian and Gumbel performance seems to already be understood in theory based on these concentration properties shown earlier in the paper. I felt that an experiment which included a model that did not satisfy completeness conditions (within the same setup as 6.2) and shows the bias of this model at the learned minimizer would be much more illuminating.

**Questions:**

- Should line 160 only consider $p \in \operatorname{ri}(\Delta)$?
- Should Theorem 3.1 include the sub-Gaussianity parameter?
- Should line 240 say "Gumbel" instead of "exponential family"?
- In Theorem 4.1, what does "whose cumulative distribution decays to zero" mean? Can you make this precise, as CDFs do not decay to zero?

---

> ### Author Response · Authors · 2024-11-20
>
> Thank you for your positive review.
> Please note that the condition on function $h_i$ does appear in Theorem 4.1's proof in Appendix 8.3. The condition that the function $h_i(\theta) = \theta_i - max_{j \ne i} \theta_j$ is an unbounded continuous function over $\Theta$ allows establishing that zero-one distributions are limit points of probabilities in the closure $\cal P$. We will make this proof idea explicit in the main text. Further, thank you for noticing that the condition on the function $h_i(\theta)$ is explicit in the proof of the completeness of Perturb-Argmax, but is missing from the conditions of Theorem 5.1. We will add this condition explicitly.
>
> Indeed, the Gumbel-Softmax is instrumental in the variational auto-encoder model, as mentioned in lines 120-121.
>
> Theorem 3.1 includes the sub-Gaussian parameter in the assumption that $\|\nabla f(\gamma) \|^2 \le \sigma^2$.
> The "whose cumulative distribution decays to zero" is an inaccurate phrasing, we meant that its density function decays to zero as $\gamma$ approaches $\pm \infty$.
> The convergence rate referenced in line 240 is indeed that of the Gumbel distribution, though we meant to convey that there are other distributions in the exponential family with an exponential convergence rate as the Gumbel.

---

> > ### Comment · Reviewer_P6qr · 2024-11-26
> > **Response to Rebuttal**
> >
> > Thank you for addressing most of my questions. I maintain my score.

---

### Official Review · Reviewer_J5XZ · 2024-11-06

**Soundness:** 2
**Presentation:** 2
**Contribution:** 2
**Rating:** 3
**Confidence:** 5

**Summary:**

The paper studies the Gumbel-max distributions in the mathematical view. The authors especially investigate the completeness, identifiability, and minimality of the distributions, i.e., the Purturb-Maxes. Finally, the advantage of Gaussian-softmax, which is an example of the Purturb-softmax, is demonstrated in the experiment section.

**Strengths:**

The paper is clearly written and well-organized.

The work is theoretically grounded.

**Weaknesses:**

Even though this is a theoretical paper, the work requires empirical evidence more than the ones suggested in the manuscript.

Also, it is not clear how the current experiments are related to the theories provided in the main body.

The proofs can be moved to the appendix.

**Questions:**

In practice, the Gumbel-softmax is more widely utilized than the Gumbel-max, and the temperature parameter is naturally added to the distribution setting. However, it seems that it is dropped in the theoretical analysis. Why is that?

---

> ### Author Response · Authors · 2024-11-20
>
> We are happy you found our work well-written and theoretically grounded.
>
> The experiments empirically demonstrate the theoretical benefits of learning with Gaussian perturbation models rather than Gumbel perturbation models. We find it hard to address the comment that the work requires more empirical evidence, as it is general and not substantial.
>
> Our framework easily extends to include the temperature scaling as in the Gumbel-Softmax models with temperature, as outlined in Equation 11. Importantly, our theoretical investigation allows us to identify the conditions under which a set of parameters $\Theta$ itself is complete. In the previous work (Maddison et al., 2017; Jang et al., 2017), completeness depends on the temperature hyper-parameter tuning.

---

> > ### Comment · Reviewer_e4WY · 2024-11-24
> >
> > > We find it hard to address the comment that the work requires more empirical evidence, as it is general and not substantial.
> >
> > I gave some specific suggestions in my review of the kinds of empirical evidence that might be helpful to being more convincing in the benefits of Gaussian-Softmax over Gumbel-Softmax in practice. Your experiments are very minimal.

---

> ### Comment · Reviewer_J5XZ · 2024-11-26
>
> After reading other reviewers' comments and the author's feedback, I decided to maintain my score.

---

### Official Review · Reviewer_e4WY · 2024-11-10

**Soundness:** 2
**Presentation:** 2
**Contribution:** 2
**Rating:** 5
**Confidence:** 3

**Summary:**

The paper studies representation properties of "Perturb-Softmax" and "Perturb-Argmax" distributions, generalizations of the Gumbel-Softmax/Gumbel-Max distributions used in a significant line of prior machine learning work on discrete random variables. They show theorems establishing general representation results, and experiments arguing that perturbing with a Gaussian rather than a Gumbel yields improved learning performance.

**Strengths:**

The idea of alternatives to the Gumbel trick is interesting. It indeed relates to FTPL (and similarly, I think, to various non-exponential mechanisms used in differential privacy), but in this context to my knowledge the Gumbel version is the only one that has been thoroughly considered.

The experimental results are promising that learning with Gaussian noise may be preferable to Gumbel noise.

**Weaknesses:**

While the setting of this paper is interesting, the presentation is a little bit disjointed:
- Your conclusion says that "Our main contribution is a theoretical study of the representation properties of Gumbel-Softmax and Gumbel-Argmax probability models." I think this is not really true; the basic derivation of Gumbel-Argmax shows that it is obviously equivalent to general softmax parameterizations of discrete distributions, in which case the completeness and minimality properties are well-known. While I don't think either of the two Gumbel-Softmax papers made explicit arguments about completeness, both included a temperature parameter, and the equivalence of Gumbel-Softmax to Gumbel-Argmax in the limit of low temperatures – explicitly stated as Proposition 1 part (c) of Maddison et al. – immediately and obviously implies completeness there. Thus most of these results were implicit in the literature already and the new ones are not _especially_ interesting in the machine learning context.
- What is quite interesting, though, is the claim that perturbing with other forms of noise can yield the same kind of completeness (and, less relevantly for learning, minimality) properties. The experiments with Gaussian-Softmax also indicate that other kinds of noise are worth exploring, although it would be nice to have a better understanding of _why_ (see discussion about the "convergence rates" argument below) or of how much improvement can be seen in a wider variety of settings. So, I think the paper should be somewhat reframed as being about the generalized Perturb-*max models, and how other choices than Gumbel can be better.
    - To this end: while the experiments here are definitely not _nothing_, they also contain no detail other than "here is the final approximation quality," and they're also not of an especially exciting scale in 2024. Ideally, I'd like to see some better poking at why optimization worked better in a simple case, e.g. one of the fixed discrete target distributions (is the parameter vector $\\theta$ to get a good approximation simpler? if not, is something about the optimization landscape simpler?). It would also be nice to have experiments for a more modern model using this as a component; just take one of the many recent models citing these papers and swap out the Gumbel.

But, before doing that, a vital issue:

- Theorem 5.1 as stated here is **plainly false**. In the theorem statement, there is no constraint on the form of random variable $\gamma$. First, they are presumably intended to be iid, but maybe this doesn't matter. So, consider using the random variable $\gamma_i$ which is a point mass at 0. Then the Perturb-Argmax distribution is $\\mathbb{E}_\\gamma[ \\mathrm{arg\\,max}\\; \\theta + \\gamma ] = \\mathrm{arg\\,max}\\; \\theta$; depending on exactly how we define argmax's behaviour in the case of ties, this can _only_ produce one-hot distributions. Since the structure of your proof is that one-hots are in the closure of $\\mathcal P$ and that this closure is convex, presumably you didn't notice this when adapting the proof of Theorem 4.1, since indeed that first property is certainly true. Presumably, then, the claim that the closure of $\\mathcal P$ is convex is false in this case. I don't know where the flaw is, but something must be wrong. In particular, since Theorem 4.1 follows a similar structure, it is particularly concerning that there is some unknown flaw that might also apply to that theorem's proof. While it's not immediately obvious to me what the problem is, I'm not so confident in how to carefully apply Rockafellar's results that you use (which are, I think, stated for $\\mathbb R^d$) to the probability simplex, and so am not confident that it's a mistake in Theorem 5.1 specifically. I cannot see arguing for anything other than rejecting the paper unless this mistake is identified and corrected, and a full correct proof provided during the rebuttal process.

Perhaps relatedly, the mathematical discussion and definition in several aspects of this paper is quite sloppy. Here are a few reasonably important ones:
- Line 195 claims that "The fundamental theorem of extreme value statistics asserts the equivalence between the softmax distribution in Equation 4 and the Gumbel-Argmax distribution in Equation 2." I don't see how this is in any way true. You only cited a 60-page lecture series which does not use this phrase to identify a theorem; presumably, you mean a [version of this theorem](https://en.wikipedia.org/wiki/Fisher%E2%80%93Tippett%E2%80%93Gnedenko_theorem), which states that the maximum of $n$ iid values can have one three asymptotic behaviours as $n \\to \\infty$, one of which is the Gumbel distribution. I don't see at all what this has to do with the distribution of a _soft_ max of a finite number of values. Luckily, though, you don't seem to use this seemingly-incorrect claim again.
- Section 3.4 "Convergence rates" compares the upper bounds on concentration of Lipschitz functions of a Gaussian variable and a Gumbel variable, and since the former is faster than the latter, vaguely imply that this means that Gaussian-Softmax is better than Gumbel-Softmax. But the purpose of the remainder of your paper is that you would use the two to identify _the exact same discrete distribution_, and thus estimating $\mathbb E\_{\\gamma \\sim g}[ f(\\theta, x, \\gamma) ]$ by Monte Carlo samples of $\\gamma$ will be _identical_ between equivalent choices of $\\theta$ with different distributions for $\\gamma$. These differences in upper bounds (which are worst-case over functions $\\gamma \\mapsto f(\\theta, x, \\gamma)$) are not relevant here.

And here are a few examples that are easy to fix, but indicative that perhaps you were not as careful in writing as you should have been:
- On line 172 you say that the probability density function of the Gumbel is $\\hat g(t) = e^{-e^{-(t+c)}}$ where $c \\approx 0.5772$. This is not true; this is the _cumulative_ density function, specifically of a Gumbel with mean $c$ and scale $1$. While scale $1$ is standard for the Gumbel-max trick, there is absolutely no reason to use mean $c$; it doesn't break anything, but by far the more typical choice would be to use mean $0$.
- Theorem 4.1 says "let $\gamma = (\\gamma\_1, \dots, \\gamma\_d)$ be a vector of random variables whose cumulative distribution decays to zero as $\\gamma$ approaches $\\pm \infty$." First, presumably you mean that the $\\gamma_i$ are iid, and it is the distribution of each of those that has the appropriate decay property. Secondly: it is trivially true of any real-valued random variable that its cumulative distribution function decays to zero as you approach $-\\infty$, and to _one_ as you approach $\\infty$. Perhaps you meant that the measure of sets $(-\\infty, -t)$ and $(t, \\infty)$ should approach zero as $t \\to \\infty$? But this is again automatically true of any real-valued variable. I actually have no idea what you mean here.
- Line 719: "A multivariate function is differentiable if its directional derivative is the same in every direction $v \\in \\mathbb R^d$, namely $\\nabla f(\\theta) = \\nabla_v f(\\theta)$ for every $v \\in \\mathbb R^d$." This is not at all true, as should be clear from the fact that your equation is equating a vector to a scalar. Rather, the directional derivatives should be _consistent_, i.e. $\\nabla f(\\theta) \cdot v = \\nabla_v f(\\theta)$ for all $v$.

Minor points of terminology:
- Your definition of identifiability is different than the one I've always heard before, and is also used e.g. [by Wikipedia](https://en.wikipedia.org/wiki/Identifiability), which is exactly what you call "minimal." I don't see much reason in giving a special name to the bijection case here, and particularly in giving one that is so widely used to mean something slightly different; I would strongly encourage you to change the terminology to "complete" (or perhaps "universal" would be more common in learning contexts) when the map from parameters to probability distributions is (almost) a surjection, and "identifiable" when it is an injection.
- Typically one would put `\appendix` at the start of the appendix, which would name the appendix A rather than 8.
- Line 713: "Convexity is a one-dimensional property." This is a strange statement; while you can define it in one dimension and extend it to other cases as you do here, there are also (many) direct definitions of convexity that work for any input vector space.

Trivial typos, etc that I happened to notice:
- line 160: you wrote $softmax$ instead of $\\mathrm{softmax}$
- "subsetset" on line 209
- Footnote 3: "dominant convergence theorem" should be "dominated convergence theorem"

**Questions:**

- How can Theorem 5.1 be corrected? Does the mistake also apply to Theorem 4.1?

- Do you have any insights as to why Gaussian-Softmax seems to be more learnable in these settings than Gumbel-Softmax? Can those insights be generalized to help guide to other noise distributions that might be even better?

---

> ### Comment · Reviewer_P6qr · 2024-11-15
> **Comment to Reviewer e4WY**
>
> On Theorem 5.1, I believe they meant to write the set of all expectations (for $\theta \in \Theta$) in between $\operatorname{ri}(\Delta)$ and $\Delta$? These components are quite important; thank you for pointing them out.

---

> > ### Comment · Reviewer_e4WY · 2024-11-15
> >
> > I’m not sure what you mean by that. I read the theorem statement as saying that for any choice of $\\gamma$ distribution, the set of distributions achievable by any $\\theta$ (these expectations) is indeed between $\\mathrm{ri}(\\Delta)$ and $\\Delta$. But when $\\gamma$ is a point mass at zero, the set does not contain $\mathrm{ri}(Delta)$; it’s only a subset of the boundary of $\Delta$, nothing in the relative interior at all!
> >
> > It might be that the proof works as long as $\\gamma$ is continuous or something like that, but I don’t know; we’ll see if the authors do when they’re ready to respond. :)

---

> ### Author Response · Authors · 2024-11-17
> **Regarding the correctness of Theorem 5.1**
>
> Respectfully, Theorem 5.1 as stated is **not false**: the theorem relies on the connection between Equation 14 $(ri(\Delta) \subseteq E_{\gamma} [
>         \arg\max(\theta + \gamma)] \subseteq \Delta)$ and Equation 13 $(\partial f(\theta) = E_{\gamma} [\arg \max(\theta+\gamma)])$. Equation 13 considers the expected argmax and sub-gradient of the expected max-value (defined in Equation 12 $(f(\theta) = E_{\gamma} [\max_{i}${$\theta_i+ \gamma_i$}])). In lines 728-729 we define the sub-gradient explicitly, which we quote here as well: A sub-gradient $p \in \partial f(\theta)$ satisfies $f(\tau) \ge f(\theta) + \langle p, \tau - \theta \rangle$ for every $\tau \in \Theta$.
>
> As a test case, let us consider the example given by Reviewer e4WY, which considers a point mass at zero. For simplicity, let's assume that $d=2$, i.e., $\theta = (\theta_1,\theta_2)$. In this case:
>
> $f(\theta)$ = $\max${$\theta_1, \theta_2 $\}
>
> $\partial f(\theta) = \arg \max$($\theta_1, \theta_2)$
>
> $(p_1,p_2) \in \partial f(\theta) $  $ \text{ iff }$  $\forall (\tau_1,\tau_2)  \max $ {$\tau_1, \tau_2 $} $\ge  \max${$\theta_1, \theta_2$}  + $\langle p, \tau - \theta \rangle$.
>
> It is clear that if $\theta_1 \ne \theta_2$ then the maximal argument is uniquely defined, and $\partial f(\theta)$ is the point-mass distribution. However, Danskin's Theorem states that when $\theta_1 = \theta_2$, then $p = (p_1,p_2)$ span the probability simplex.
>
> To make this surprising result more intuitive, let's consider the case when $\theta_1 > \theta_2$. In this case $\max${$\theta_1,\theta_2$} $= \theta_1$ and the point-mass $p = (1,0)$ is the (sub-)gradient. To see that, one can verify that
> $\max$ {$\tau_1, \tau_2$}  $\ge \tau_1 = \theta_1  + (\tau_1 - \theta_1)  = \theta_1  + \langle (1,0), \tau - \theta \rangle  $.
>
> However, when $\theta_1 = \theta_2$, for every $p = (p_1,p_2)$ that is a probability distribution (i.e., non-negative and sum up to unity) the following two equations hold:
>
> $\max${$\tau_1, \tau_2 $} $\ge \langle p, \tau \rangle$
>
> $\max${$\theta_1, \theta_2 $} $= \langle p, \theta \rangle$
>
> and combining these two equations we get that any $p = (p_1,p_2)$ that is a probability distribution is also a sub-gradient, i.e., $p \in \partial f(\theta)$:
>
> $\max${$\tau_1, \tau_2$} $ \ge \langle p, \tau \rangle = \max${$\theta_1, \theta_2$}$ +  \langle p, \tau \rangle -  \langle p, \theta \rangle $.
>
> The general setting is described by [Danskin's theorem](https://en.wikipedia.org/wiki/Danskin%27s_theorem), where the compact set is the discrete set {$1,...,d$}. One can verify it by following the example in Sec 4.5, page 247 of the book Convex Analysis and Optimization, by Bertsekas and Ozdagler, 2003.
>
> We also refer to Proposition 5.5 in the paper for an analysis of the Perturb-Argmax probability distribution when considering the case of $\Theta = \mathbb{R}^2$ and $\gamma = (\gamma_1,\gamma_2)$ a vector of uniformly distributed discrete random variables. Concretely, as illustrated in Figure 3, the sub-differential of the expected max value spans the probability simplex.

---

> ### Comment · Reviewer_e4WY · 2024-11-20
>
> I of course agree that the entire probability simplex is a subgradient of the argmax when the $\\theta$s are equal; this is incontrovertible and probably the most common case of subgradients that aren't gradients. (In retrospect, this should have been clear that this is where it came from.)
>
> This abstraction about the connection to duality, though, I think obscures a very important point about the interpretation of the argmax distributions. Let's consider $\gamma$ a point mass at zero in the two-dimensional case. Then, $\\{ \\mathrm{arg\\,max} \\, \\theta : \\theta \\in \\mathbb R^2 \\}$ can contain three kinds of settings: $\\theta\_1 > \\theta\_2$ (a point mass on variable 1), $\\theta\_1 < \\theta\_2$ (a point mass on variable 2), or $\\theta\_1 = \\theta\_2$ (not clearly defined). For the conclusion of your theorem to be true, the argmax operator must break ties not either according to some fixed rule (which would give only a point mass) or uniformly (which would give a uniform distribution over the two variables), but with _arbitrary probabilities_ that are not specified anywhere in the model. This is not reasonable.
>
> That said, I'm now more satisfied that if you simply require the $\gamma$ distributions to be nonatomic, then I have no reason to doubt either theorem.

---

> ### Author Response · Authors · 2024-11-20
> **Regarding the discussion, definitions and conntribution**
>
> Regarding the mathematical discussion and definition:
>
> 1. Line 195 and the fundamental theorem of extreme value statistics.
> While not explicitly phrased, we attribute understanding of the equivalence between the softmax distribution and the Gumbel-Argmax to Julius Gumbel, hence we referred to the 1954 notes. We agree that Fisher and Tippett have had the same understanding and we readily agree their work should be cited. Indeed, the same understanding also appears in Choice Theory literature (for example, Luce, 1959). Our goal was to reference the origin of the theory, though not Fisher and Tippett, nor Gumbel have cited this theorem explicitly. We will also cite Luce's work for completeness.
>
> 2. A convergence rates analysis motivates our study of the Gaussian-Softmax and Gumbel-Softmax distributions' representation properties. Their different convergence rates and equivalent representation properties justify fitting probabilities with Gaussian instead of Gumbel random perturbations. The faster convergence of Gaussian-Softmax may hint at their efficient statistical nature that emerges from our experiments (i.e., their improved convergence when using the same number of samples.)
>
> We agree that our main contribution is not primarily the theoretical study of the representation properties of Gumbel-Softmax and Gumbel-Argmax probability models. A better phrasing would be that our framework for investigating the representation properties of these models allows for extending it to identify the representation properties for a wide range of perturbation models. As for the completeness property of Gumbel-Softmax arising in the zero temperature limit in previous work (as mentioned by the reviewer) --- it is an insightful comment that we will add to the work and thank the reviewer for it. Our theoretical investigation also allows identifying the conditions under which a set of parameters $\Theta$ itself is complete, rather than a property achieved by a convergence argument.
>
> Regarding "easy-to-fix" inaccuracies. Thank you for spotting these inaccuracies, we will surely correct them.

---

> ### Comment · Reviewer_e4WY · 2024-11-24
>
> Point number 1: I wasn't asking so much about the citation as the fact that I don't see what the fundamental theorem of extreme value statistics has to do with the claim you make here. I still don't; if it were true that "the representation properties of completeness and minimality of the softmax operation are identical to the properties of the Gumbel-Argmax probability distribution", why would the rest of your paper be necessary?
>
>
> Point number 2: Having thought about this more, I think we're both right. :)
>
> Let's think first about Perturb-Argmax; I'll change your notation to make things clearer. Here, you want to estimate the mean of a function $h(n)$ where $n$ is the integer from your discrete distribution, obtained as $n \in \operatorname{argmax} \theta + \gamma$; call this $n(\theta, \gamma)$, so the overall problem is $$\min\_\theta \operatorname*{\mathbb{E}}\_{x \sim D}\left[ \operatorname*{\mathbb{E}}\_{\gamma \sim g} h(x, n(\theta, \gamma)) \right] = \min\_\theta \operatorname*{\mathbb{E}}\_{x \sim D}\left[ \operatorname*{\mathbb{E}}\_{n \sim p_{\theta,g}} \left[ h(x, n) \right] \right].$$
> In practice, this would be estimated with Monte Carlo samples over both $x$ and $n$. Thus, for any $\theta, g$ and $\theta', g'$ such that the corresponding distributions over $\mathbb N$ are the same, the Monte Carlo convergence rate will be identical; it doesn't matter at all if this is based on Gaussians, Gumbels, or whatever else. This is the case I was thinking about in my initial review.
>
> When using Perturb-Softmax, rather than obtaining a "hard" sample $n$, you get a "soft" sample of probabilities $\hat p = \operatorname{softmax}(\theta + \gamma)$. Here, equivalence between distributions means that $\mathbb{E}\_\gamma \hat p$ is the same, but the distributions of $\hat p$ between equivalent, say, Gumbel-Softmax and Gaussian-Softmax distributions will be different. Since when using temperature 1 softmax is 1-Lipschitz, indeed we have that the function $\gamma \mapsto \operatorname{softmax}(\theta + \gamma)$ is also 1-Lipschitz, and so when we want to estimate the mean of some function of $\hat p$, the convergence rates can indeed be meaningfully different in terms of the Lipschitz properties of $h(\hat p)$. That the worst-case bound over $h$ is sub-exponential rather than sub-gaussian is indeed suggestive. It would be nice to have a more thorough study of this, though: in the cases where Gaussian-Softmax performs better than Gumbel-Softmax, is this indeed what's happening? I *expect* (but it should be confirmed) that samples from Gumbel-Softmax have $\hat p$ "closer" to one-hot, e.g. lower-entropy, than Gaussian-Softmax; it is reasonably intuitive that this would give slower-converging Monte Carlo behavior, but this is something that I think your paper needs to study more carefully.

---

### Meta-Review · Area_Chair_vRDf · 2024-12-18

**Metareview:**

This work investigates the representation properties of the Gumbel-Softmax and Gumbel-Argmax probability distributions, which are used for learning discrete tokens and structures in generative and discriminative models, respectively. The study identifies the conditions under which these distributions are complete (able to represent any probability distribution) and minimal (able to represent a distribution uniquely), using convexity and differentiability. The analysis extends to general probability models like Perturb-Softmax and Perturb-Argmax, concluding that certain parameter sets allow for complete and minimal representations, with experimental results validating the faster convergence of Gaussian-Softmax compared to Gumbel-Softmax.

The problem addressed by the paper is clearly important and practical.  Although the paper is meant to be theoretical, the experimental results are really minimal and lack some important comparisons as the reviewers have pointed out.  There are also a few concerns raised by Reviewer e4WY that remain not fully satisfactorily addressed in the rebuttal.  So the paper needs one more round of revision.

**Additional Comments On Reviewer Discussion:**

The rebuttal has been noted by the reviewers and have been taken into account by the AC in the recommendation of acceptance/rejection.

---

### Decision · Program_Chairs · 2025-01-22

Reject